# MANF antagonizes nucleotide exchange by the endoplasmic reticulum chaperone BiP

Yahui Yan [1], Claudia Rato [1], Lukas Rohland [1,2], Steffen Preissler [1] & David Ron [1]

Despite its known role as a secreted neuroprotectant, much of the mesencephalic astrocyte-derived neurotrophic factor (MANF) is retained in the endoplasmic reticulum (ER) of producer cells. There, by unknown mechanisms, MANF plays a role in protein folding homeostasis in complex with the ER-localized Hsp70 chaperone BiP. Here we report that the SAF-A/B, Acinus, and PIAS (SAP) domain of MANF selectively associates with the nucleotide binding domain (NBD) of ADP-bound BiP. In crystal structures the SAP domain engages the cleft between NBD subdomains Ia and IIa, stabilizing the ADP-bound conformation and clashing with the interdomain linker that occupies this site in ATP-bound BiP. MANF inhibits both ADP release from BiP and ATP binding to BiP, and thereby client release. Cells lacking MANF have fewer ER stress-induced BiP-containing high molecular weight complexes. These findings suggest that MANF contributes to protein folding homeostasis as a nucleotide exchange inhibitor that stabilizes certain BiP-client complexes.

---

[1] Cambridge Institute for Medical Research, University of Cambridge, Cambridge CB2 0XY, UK. [2] Present address: Center for Molecular Biology (ZMBH) of Heidelberg University, Heidelberg, Germany. These authors contributed equally: Yahui Yan, Claudia Rato. Correspondence and requests for materials should be addressed to S.P. (email: sp693@cam.ac.uk) or to D.R. (email: dr360@medschl.cam.ac.uk)

The protein known as MANF was first characterized functionally as an agent in the supernatant of a rat astrocyte cell line that protected cultured dopaminergic neurons from death[1]. While an extensive literature addresses the role of MANF as a secreted molecule exerting non-cell-autonomous effects (reviewed in ref. [2]), other observations point to an intracellular function for MANF, specifically in protein-folding homeostasis in the ER.

MANF's N-terminus contains a cleavable signal sequence, typical of proteins that enter the secretory pathway. However, unlike most secreted proteins, MANF ends with a conserved C-terminal RTDL sequence, well suited to engage the KDEL receptor and promote ER retention[3]. The *MANF* gene is prominently induced in the course of the unfolded protein response (UPR)[4] and together with few known ER quality control factors, MANF is induced by overexpression of misfolding-prone secreted proteins[5]. Furthermore, disruption of *MANF* gene function leads to enhanced activity of UPR markers in cultured cells[6] and in the tissues of *MANF* knockout mice[7] and worms[8]. Together, these observations hint at MANF's role in the adaptation of cells to the stress imposed by enhanced levels of unfolded ER proteins.

The ER-localized Hsp70 chaperone BiP plays an important role in protein-folding homeostasis. Like Hsp70s in other compartments, BiP does so by the reversible binding and release of unfolded client proteins, a tightly regulated process that depends on the concentration of active BiP and on the nucleotide bound to it. In the ATP-bound state, BiP exchanges clients with high on and off rates. However, J-domain co-chaperones specify BiP–client protein interactions by triggering the hydrolysis of ATP in association with the client. In its ADP-bound form, BiP binds clients stably. A different class of co-chaperones, the nucleotide exchange factors (NEFs), promote completion of the chaperone cycle by directing the turnover of the BiP–client complex through accelerated exchange of the bound nucleotide from ADP to ATP. Cytosolic Hsp70 chaperones are subjected to an additional layer of regulation imposed by Hip, a protein that antagonizes nucleotide exchange and thereby stabilizes certain chaperone–client interactions[9]. However, a counterpart nucleotide exchange inhibitor (NEI) activity in the ER has not, to date, been reported.

Given the importance of factors that interact with BiP and regulate its chaperone cycle, activity, and abundance, we were intrigued by the observation of a physical interaction between MANF and BiP in cultured human cells[10] and by evidence for genetic interactions between their encoding genes in flies[11]. Here, we report on a structural and biochemical characterization of that interaction. Our studies suggest that MANF contributes to protein-folding homeostasis in the ER by antagonizing nucleotide exchange on BiP, thus stabilizing certain BiP–client interactions.

## Results

**MANF interacts with BiP's nucleotide-binding domain**. To search for a role for MANF in protein-folding homeostasis in the ER, we took advantage of CHO-K1 S21 cells. These cells have stably integrated reporter genes for the UPR; *CHOP:GFP* reports on the PERK branch of the UPR and *XBP1s:Turquoise* reports on the IRE1 branch[12]. The *MANF* gene was inactivated by CRISPR-Cas9 genome editing, resulting in *MANF* nullizygous clones (Fig. 1a, b). Consistent with previous observations made in HeLa cells[6] or tissues of knockout animals[7,8], MANF-deficient CHO-K1 cells also had basally heightened activity of their UPR markers (Fig. 1c), which was suppressed to wild-type levels by rescue of the mutation with a retrovirus encoding MANF (Supplementary Fig. 1a, b).

ER calcium depletion has been associated with enhanced secretion of MANF protein. However, even under such conditions, a substantial pool of MANF is found in cell lysates[10]. This feature was conserved in CHO-K1 cells (Fig. 1d and Supplementary Fig. 1c) and suggested that, in addition to whatever role secreted MANF might have as a factor involved in intercellular communication, an intracellular role for MANF should also be considered.

MANF has been reported to associate with the ER chaperone BiP[10]. In our hands, too, BiP was conspicuously recovered in complex with MANF from cell lysates after in vivo cross-linking (Fig. 2a). Both BiP and MANF are bipartite proteins (Fig. 2b). To further characterize their physical interaction, we immobilized biotinylated MANF, its N-terminal SAPLIP (Saposin-like protein) or its C-terminal SAP (SAF-A/B, Acinus, and PIAS) domains onto a Bio-layer interferometry (BLI) probe and measured the optical interference signal created by BiP binding. A binding signal was registered on MANF and SAP probes when BiP was in an ADP-containing solution, but not by ATP-bound BiP. BiP did not interact with the SAPLIP probe in either nucleotide-bound states (Fig. 2c). The binding signal of BiP-ADP with either MANF or the isolated SAP domain was conspicuously bi-phasic, with a phase of rapid association and rapid dissociation and second phase of slow association and dissociation. The fast phase was a highly reproducible observation that was also recapitulated by the interaction of MANF (and SAP, but not SAPLIP) with BiP's isolated nucleotide-binding domain (NBD) (Fig. 2d and Supplementary Fig. 2a). The slow phase of binding between BiP and MANF or SAP was more variable and presently remains unclear in origin. Neither MANF nor SAP bound the isolated BiP substrate (client)-binding domain (SBD, Fig. 2d). The BiP used in the binding experiments contained a V461F mutation (BiP$^{V461F}$) that enfeebles client binding by Hsp70 proteins[13,14]. Therefore, we regard it as unlikely that engagement of MANF as a conventional BiP client made an important contribution to the binding signal.

MANF and SAP binding to BiP's NBD was nucleotide-independent, consistent with the observation that the isolated NBD of Hsp70s is locked in the conformation it assumes in the ADP-bound intact chaperone, regardless of the presence or identity of the bound nucleotide[15,16]. The fast association and dissociation kinetics of MANF (and SAP) with BiP NBD and the limited time resolution of BLI precluded direct measurement of the $k_{on}$ and $k_{off}$, however, the steady-state binding signal was saturable with a $K_{1/2\ max}$ of 10–15 μM (Fig. 2e, Supplementary Fig. 2b, c). A dissociation constant in the micromolar range is typical of the interactions between Hsp70 chaperones and their co-regulators[17–19]. Thus, the estimates of the affinity of their interaction are consistent with the idea that MANF might regulate some aspect of BiP function.

**Structural insights into the BiP–MANF complex**. To better understand the details of the interaction between MANF and BiP, we co-crystallized BiP's NBD in complex with the isolated SAP domain or with MANF. The structure of the NBD–SAP complex was solved at 1.57 Å resolution by molecular replacement using BiP NBD (PDB 3LDN) as a search model. The 2.49 Å resolution NBD–MANF complex was solved by molecular replacement using the aforementioned NBD–SAP structure and the SAPLIP domain from a MANF structure (PDB 2W51) as search models. In both complexes, the SAP domain assumes the same structure, which is furthermore similar to that previously observed in crystallographic or NMR studies of isolated MANF[20–22]. The compact SAP domain docks against the cleft between BiP's Ia and IIa NBD subdomains (Fig. 3a).

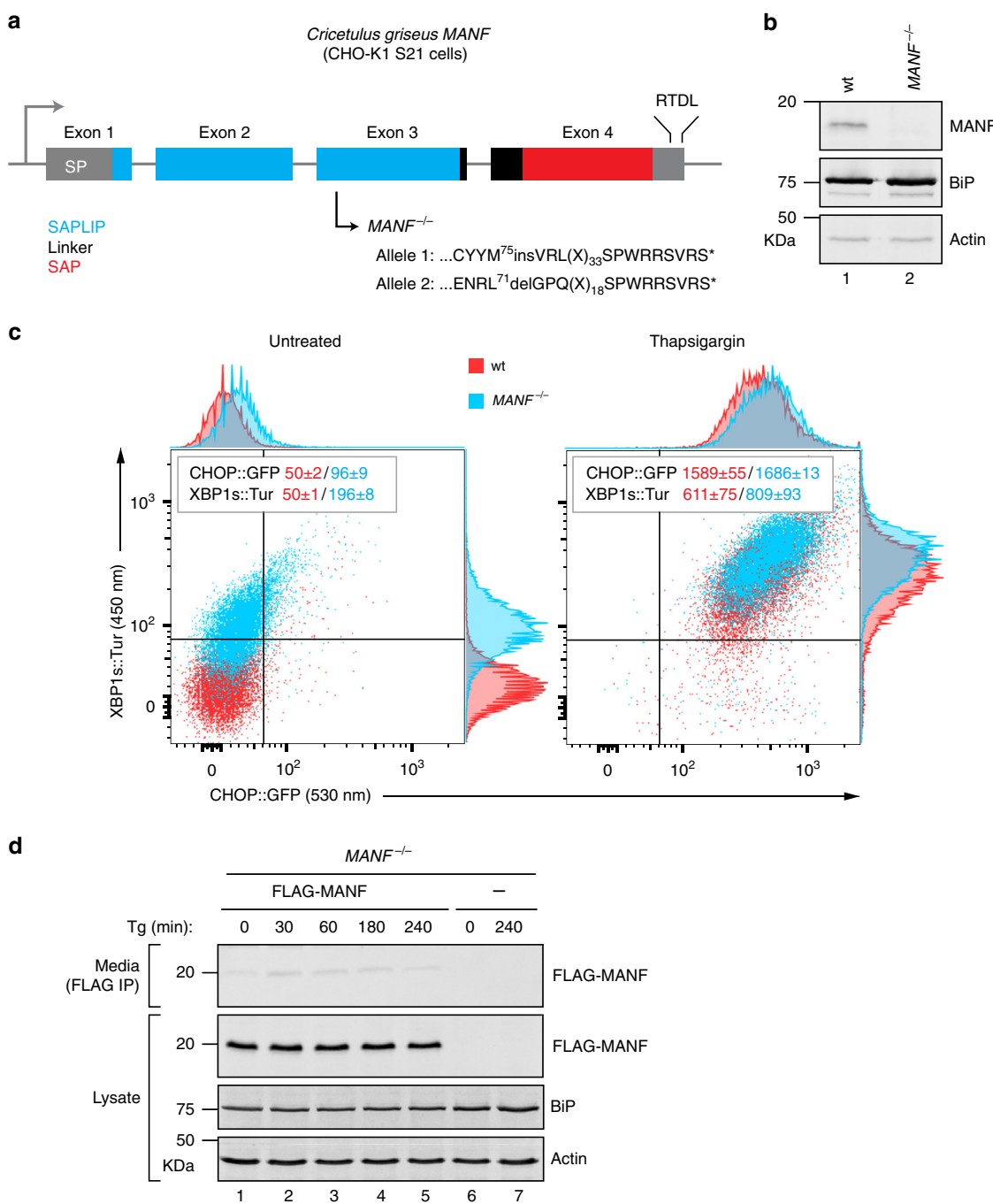

**Fig. 1** A heightened UPR in *MANF* knockout cells. **a** Schematic illustration of the CHO-K1 *MANF* gene. The encoded N-terminal SAPLIP (Saposin-like protein; blue) and the C-terminal SAP (SAF-A/B, Acinus, and PIAS; red) domains as well as the signal peptide (SP), linker region (black), and RTDL motif are shown. The encoded amino acid sequence surrounding the mutations (caused by CRISPR-Cas9-mediated nucleotide insertion or deletion) are noted for each allele. Both mutations result in premature termination of translation interrupting the SAPLIP domain and deleting the SAP domain. **b** Immunoblot of MANF in lysates of CHO-K1 S21 wild-type (wt) and *MANF⁻/⁻* cells. The BiP and actin signal in the samples is also shown. This experiment has been reproduced independently three times. Note that no MANF signal was detected in the sample from *MANF⁻/⁻* cells lysate. **c** Flow cytometry plots of *CHOP: GFP* and *XBP1s:Turquoise* UPR reporters in untreated and thapsigargin-treated (16 h) CHO-K1 S21 wild-type and *MANF⁻/⁻* cells. The inset shows the median ± SD of the GFP and Turquoise fluorescence signals of the wt (red) and *MANF⁻/⁻* (blue) cells from three independent experiments (similar results were obtained with two independently derived *MANF⁻/⁻* clones). **d** Immunoblot of FLAG-tagged MANF from cell lysates and FLAG-M1-immunoaffinity purified proteins (FLAG IP) from the corresponding cell culture supernatants (media) of parental CHO-K1 S21 *MANF⁻/⁻* cells and cells stably expressing FLAG-M1-MANF. Cells were untreated or treated with thapsigargin (Tg; 0.5 μM) for the indicated times. The content of BiP and actin in the samples is provided as a loading control. Uncropped images for panels (**b**) and (**d**) and source data for panel (**c**) are provided as a Source Data file

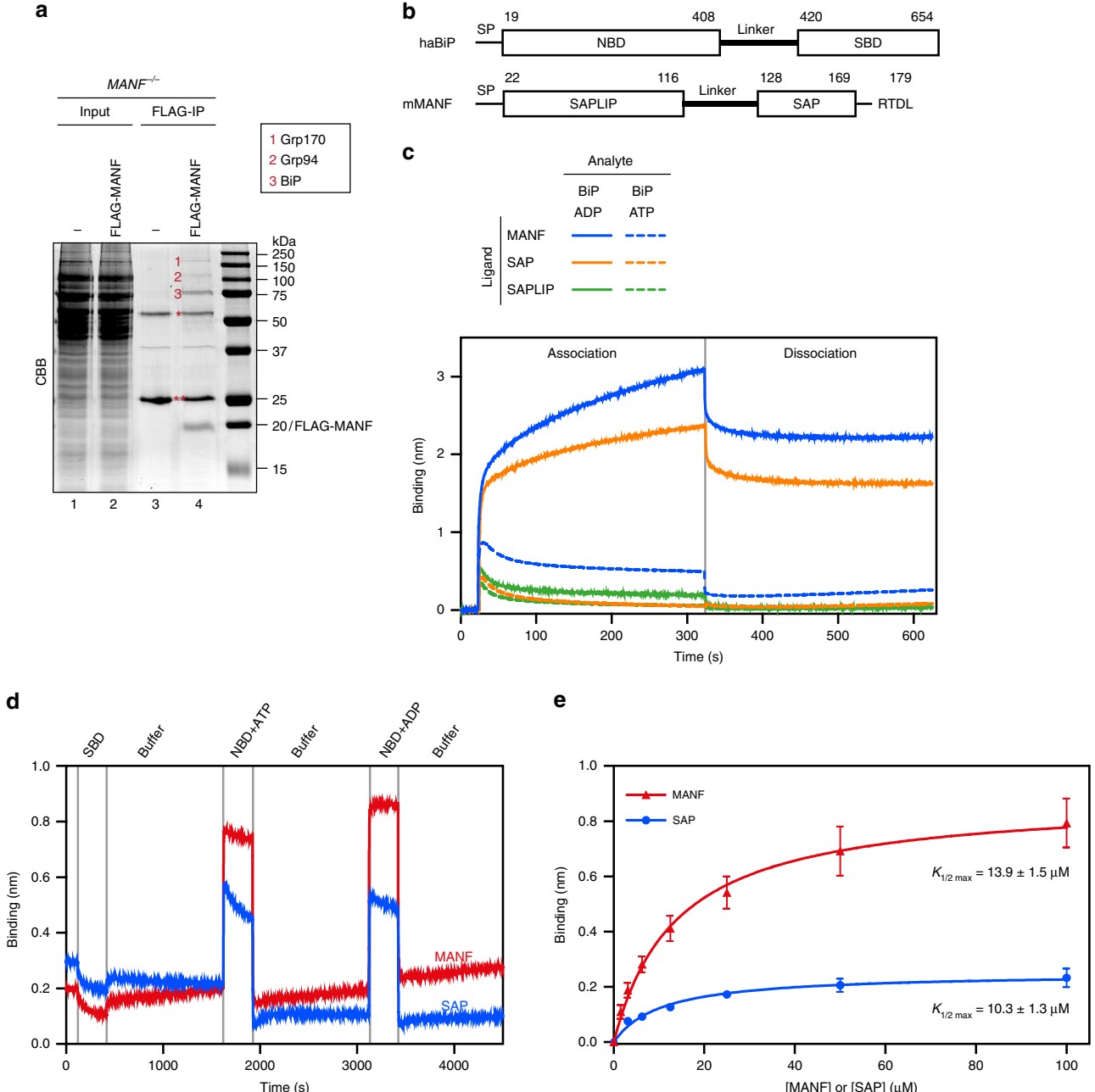

**Fig. 2** MANF associates with the ER chaperone BiP in vivo and in vitro. **a** Coomassie-stained (CBB) SDS-PAGE gel of lysates (Input) and FLAG-M1-immunoaffinity purified proteins (FLAG-IP) from parental CHO-K1 S21 $MANF^{-/-}$ cells and cells stably expressing FLAG-M1-MANF. Indicated bands (1–3) of proteins recovered in complex with FLAG-M1-MANF were individually excised and analyzed by mass spectrometry. The heavy (one asterisk) and light (two asterisks) chains from the ANTI-FLAG M1 agarose affinity gel are indicated. Data representative of two independent experiments are shown. Note that BiP (72 kDa) was unambiguously identified in band 3. **b** Schema of the domain structure of Chinese hamster BiP (haBiP) and mouse MANF (mMANF). **c** Bio-layer interferometry (BLI) signals of streptavidin biosensors loaded with biotinylated MANF, the isolated SAP or SAPLIP domains exposed to $BiP^{T229A-V461F}$ (46 μM) in the presence of 2 mM ADP or ATP. Note that SAPLIP does not bind BiP, while MANF and SAP preferentially interact with BiP in its ADP state. **d** BLI signals of streptavidin biosensors loaded with biotinylated MANF or the isolated SAP domain sequentially exposed to BiP SBD or NBD in presence of ATP or ADP (the trace of the biosensor loaded with SAPLIP is shown in Supplementary Fig. 2a). **e** Steady-state analysis for the binding affinity of BiP NBD to MANF or the isolated SAP domain. Streptavidin biosensors were loaded with biotinylated BiP NBD and exposed to MANF or SAP at the indicated concentrations. BLI signals at equilibrium were plotted against the concentrations and fitted. Data points represent mean values and SD bars from three independent experiments. Representative raw data traces are shown in Supplementary Fig. 2b, c

These structures were obtained with the isolated NBD, which is locked in the ADP-like state, however, superposition of BiP's structure in the ATP-bound conformation reveals that MANF binding to the NBD is incompatible with docking of BiP's interdomain linker in the Ia–IIa cleft (Fig. 3b and Supplementary Movie 1), which is a key feature of the ATP-bound state of Hsp70 chaperones[23–25]. By contrast, superposition of a structure of the related bacterial DnaK in the ADP-bound state, in which the NBD and SBD are undocked, suggests that the SAP domain could engage the NBD of a client-bound Hsp70 (Fig. 3c).

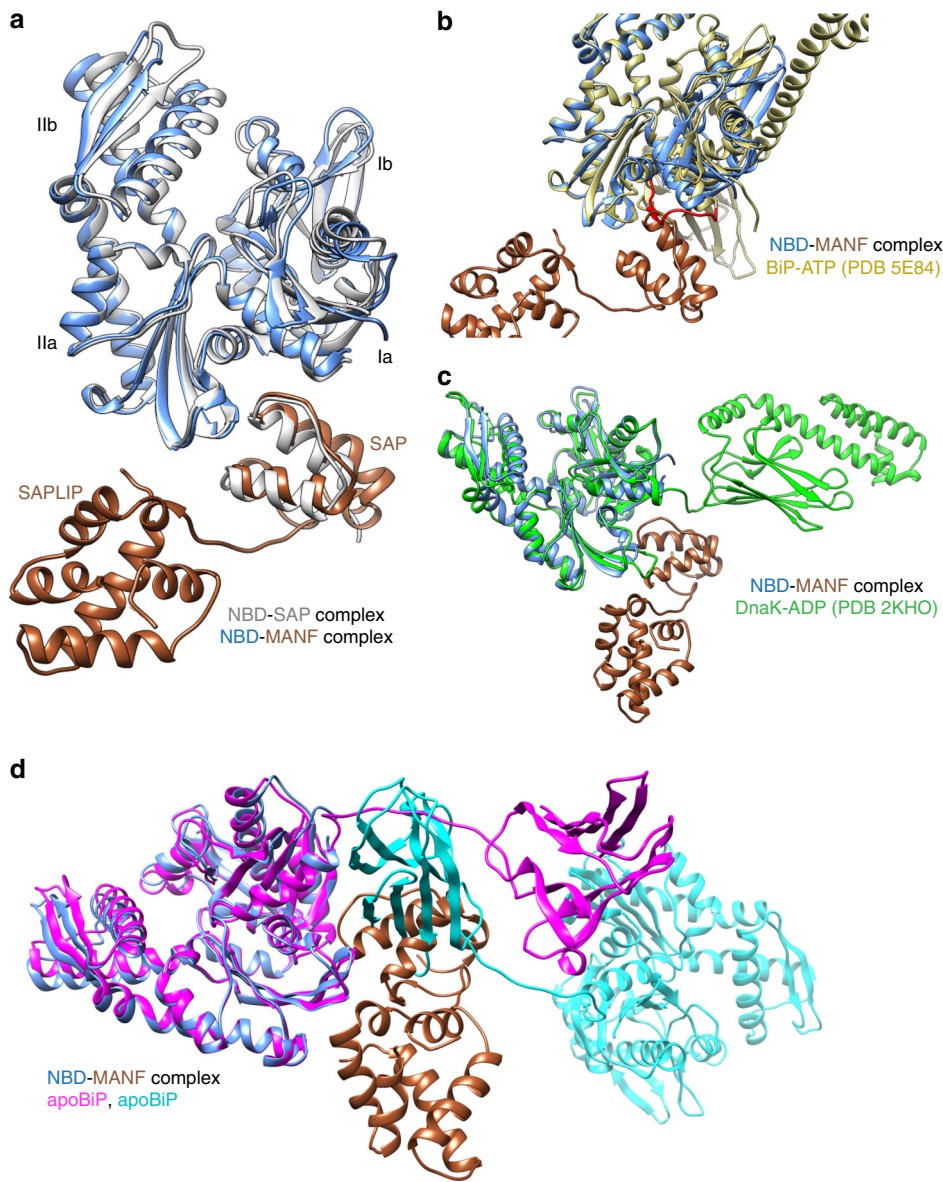

**Fig. 3** Crystal structure of BiP NBD in complex with MANF or SAP. **a** Superimposed structures of BiP NBD in complex with MANF or the isolated SAP domain (silver) showing the same binding interface between SAP and NBD (RMSD = 1.3 Å over 379 Cα atoms). The NBD–SAP complex is colored in gray, while MANF is gold and its bound NBD is blue. **b** Overlay of the NBD–MANF complex (from **a**) with full-length BiP (in the ATP state, yellow; PDB 5E84) aligned by their NBD. Note the steric clashes between SAP and the BiP interdomain linker (colored red). **c** Overlay of the NBD–MANF complex (from **a**) and one state of the solution structure of DnaK in the apo/ADP conformation (green; PDB 2KHO) aligned by their NBD. Note that in the domain-undocked (apo/ADP) state, Hsp70 (DnaK) accommodates the binding of MANF to its NBD. **d** Overlay of the NBD–MANF complex (from **a**) onto the structure of apo BiP[V461F]. MANF (gold) can be accommodated by an individual BiP molecule (magenta) in the extended, domain-undocked conformation, while a significant steric clash is noted with a crystallographic symmetry-related BiP molecule (cyan), which contacts the interdomain linker of the former BiP (magenta)

Attempts to co-crystalize MANF in complex with BiP in a conformation consistent with client binding were unsuccessful. Nonetheless, the BiP component of the crystallization mix yielded informative crystals (at a resolution of 2.1 Å). Despite a truncation of the C-terminal α-helical lid (28–549) and the presence of a V461F mutation (known to inhibit client binding), in the crystal lattice, BiP[V461F] molecules were packed with the hydrophobic interdomain linker of one protomer engaged as a conventional client in the SBD of an adjacent protomer (Supplementary Fig. 3a). A similar daisy-chain arrangement had previously been observed in the crystal structure of a bacterial Hsp70 in the ADP-bound conformation[26]. Genetic and biochemical evidence suggests that BiP oligomers, which are known

to occur both in vitro[27] and in cells[28], assume a similar architecture[29]. Interestingly, superposition of the NBD–MANF complex onto the structure of BiP oligomers reveals a clash with the SBD domain of the adjacent protomer that engages the interdomain linker (Fig. 3d). These crystallographic findings fit with MANF's observed preference for the ADP-bound state of BiP, but suggest that MANF would be excluded from the core of BiP oligomers. MANF engages BiP at a surface that is also predicted to be contacted by J-domain proteins[30]. However, unlike J-domain proteins, MANF does not significantly increase BiP's ATPase activity (Supplementary Fig. 3b).

Charge complementarity between surface residues appears to play an important role in stabilizing the NBD–MANF complex.

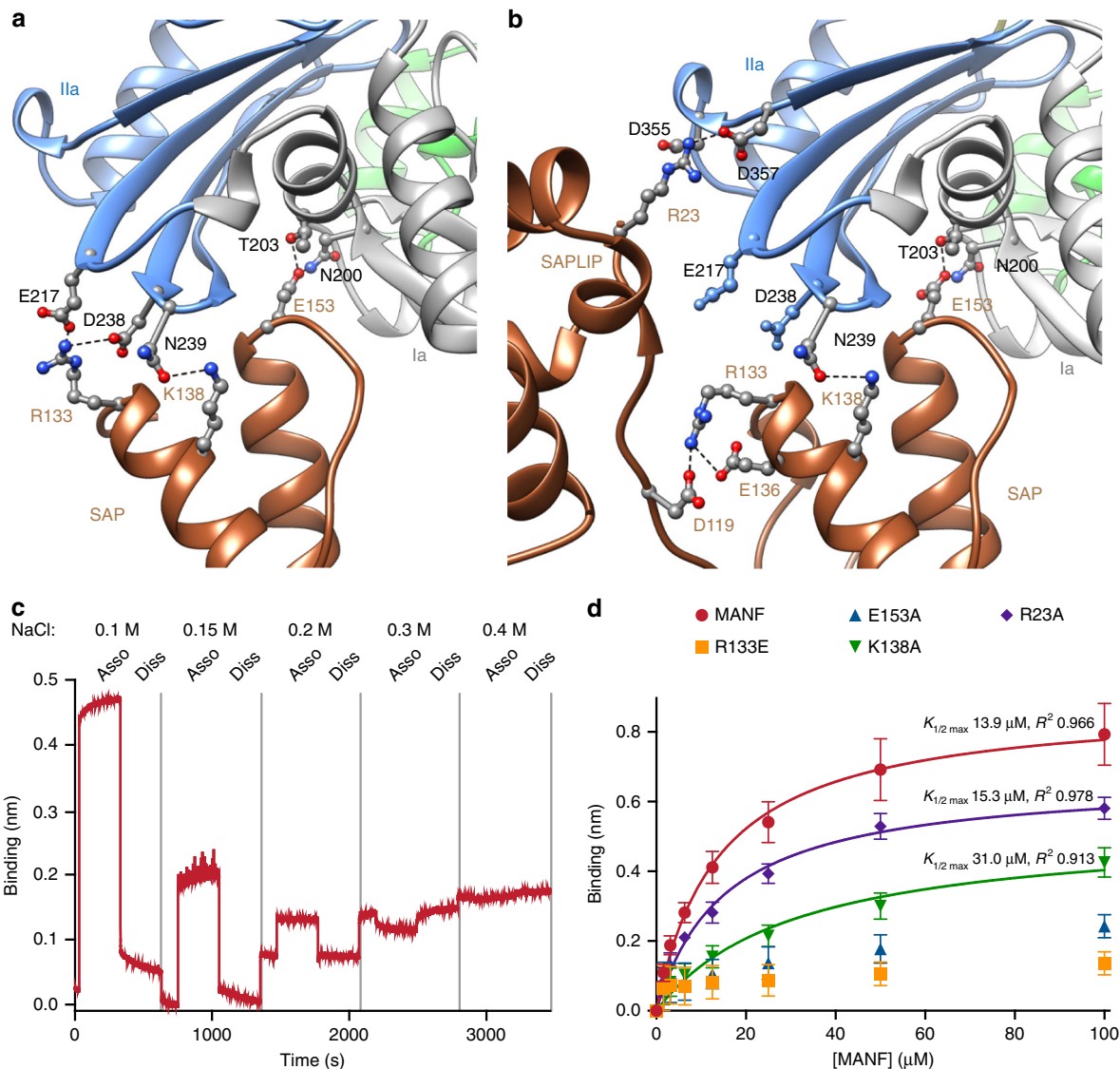

**Fig. 4** Electrostatic interactions at the interface of MANF and BiP NBD. **a** Detailed view of the NBD–SAP interface highlighting R133, K138, and E153 of SAP (gold) that form ionic interactions and hydrogen bonds with indicated residues of BiP NBD subdomains IIa (blue) and Ia (gray). **b** Detailed view of the NBD–MANF complex structure colored as in (**a**). **c** Bio-layer interferometry (BLI) trace (association and dissociation phases) of the binding signal arising from the interaction of streptavidin BLI biosensors loaded with biotinylated BiP NBD and exposed to MANF (at 10 μM) in buffer solutions containing the indicated salt concentrations. Shown is a representative of three experiments. **d** Plot of the binding signal at equilibrium arising from the interaction of streptavidin BLI biosensors loaded with biotinylated BiP NBD and the indicated concentrations of wild-type MANF or its mutants. Data points represent mean values and SD bars from three independent experiments. The $K_{1/2\ max}$ and $R^2$ values as well as fit lines are provided for the wild-type MANF and the R23A and K138A mutants. $K_{1/2\ max}$ and $R^2$ values are not provided for the R133E and E153A mutants as these gave rise to binding signals that were too weak to fit to a saturable one-site binding model. Source data are provided as a Source Data file

MANF$^{E153}$ hydrogen bonds with BiP$^{N200}$ and BiP$^{T203}$, and MANF$^{K138}$ bonds with BiP$^{N239}$, contacts observed in both the NBD–SAP and the NBD–MANF structures (Fig. 4a, b). MANF$^{R133}$ undergoes different interactions in the two structures, forming interchain bonds with BiP$^{E217}$ and BiP$^{D238}$ in the NBD–SAP crystal (Fig. 4a), and intrachain bonds with MANF$^{D119}$ and MANF$^{E136}$ in the BiP–MANF complex (Fig. 4b). MANF$^{R23}$ is observed to engage BiP$^{D355}$ and BiP$^{D357}$ in a potentially stabilizing network of hydrogen bonds (Fig. 4b). The importance of ionic interactions to the stability of the BiP–MANF complex is supported by its salt sensitivity, as reflected in the attenuation of the BLI signal upon increasing salt concentration (Fig. 4c), and by the effect of mutation of individual residues in MANF on the BLI signal arising from the interaction of an

immobilized BiP NBD with solutions of wild-type and mutant MANF (Fig. 4d).

**MANF binding attenuates nucleotide exchange on BiP.** Selective association of MANF with the ADP-bound conformation of BiP suggested the possibility that MANF might stabilize the bound nucleotide. Nucleotide release from Hsp70 chaperones can be measured by following the quenching of fluorescence, as bound MABA-ADP is released into the aqueous environment in the presence of excess non-fluorescent nucleotide (to prevent re-binding)[31]. Both the presence of MANF or the isolated SAP domain inhibited ADP release (Fig. 5a). The SAPLIP domain and mutant MANF$^{R133E}$ and MANF$^{E153A}$ were inactive, whether assayed with the isolated NBD (Fig. 5b) or with intact BiP and

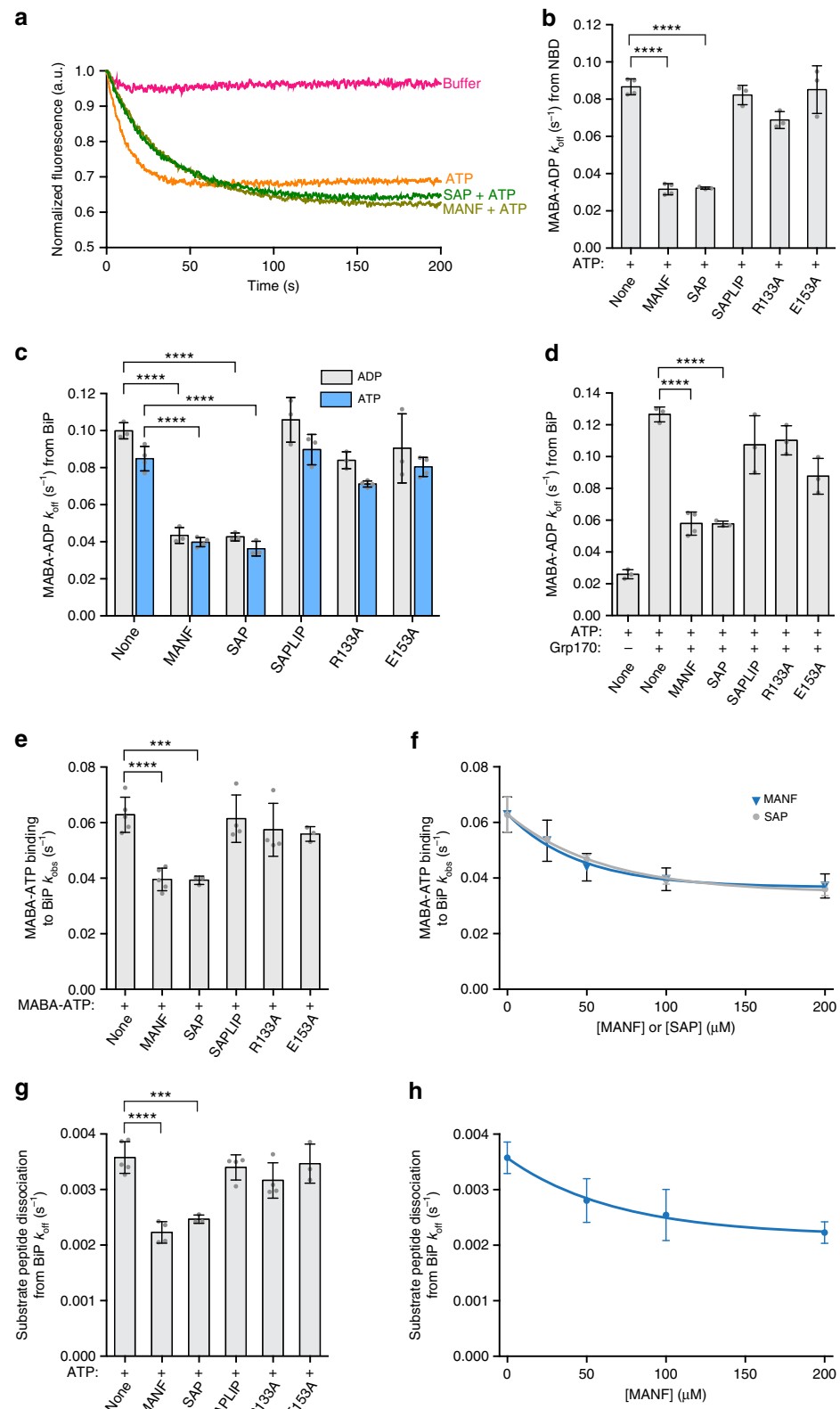

with either unlabeled ADP or ATP as the competitor (Fig. 5c). The inhibitory effect was concentration-dependent (Supplementary Fig. 4a, b), and the $IC_{50}$ of 26.5 μM for MANF and 32.4 μM for the isolated SAP domain were in the range of the apparent dissociation constants of their binding to BiP's NBD, as estimated from the BLI experiments (Fig. 2e).

MANF's inhibitory effect on nucleotide release was also observed in the presence of the nucleotide exchange factor Grp170 and physiological concentrations of calcium (Fig. 5d). In this scenario, the presence of MANF was able to antagonize more than half of the stimulatory effect of Grp170 on nucleotide release, pointing to the potential functional importance of

**Fig. 5** MANF inhibits nucleotide exchange and ATP-induced substrate release from BiP. **a** Representative plot of fluorescence against time of pre-formed complexes of MABA-ADP and BiP NBD (1.25 μM) challenged at t = 0 with buffer solution containing ATP (125 μM) and MANF or its SAP domain (200 μM). **b** Bar diagram of MABA-ADP release rates from BiP NBD in the presence of MANF, the isolated SAP or SAPLIP domain or the indicated mutant forms of MANF. Bars represent mean values ± SD from three to five independent experiments (****$P<0.0001$, unpaired Student's $t$ test). **c** As in (**b**) but measuring the MABA-ADP release from full-length BiP in the presence of either ADP or ATP as the competitor. **d** As in (**b**) but with additional presence of the NEF Grp170 (1.25 μM) and physiological concentrations of calcium (0.5 mM). **e** Bar diagram of rate of MABA-ATP loading onto BiP in the absence or presence of the indicated proteins (final: 1.25 μM BiP, 1.25 μM MABA-ATP, 100 μM MANF or mutants). Bars represent mean values ± SD from three independent experiments (***$P<0.001$ and ****$P<0.0001$, unpaired Student's $t$ test). **f** Plot of $k_{obs}$ for binding of MABA-ATP to BiP (as in **e**) against final concentration of MANF or SAP. The mean values and SD bars of three independent experiments are plotted. Single exponential best fit lines are shown. **g** Bar diagram of $k_{off}$ of fluorescently bound substrate peptide (peptide-LY) from BiP. BiP–substrate complexes were pre-formed in presence of ADP and dissociation of pre-formed BiP–peptide complexes was induced by the introduction of ATP in the absence or presence of MANF derivatives (200 μM). Bars represent mean values ± SD from three independent experiments (***$P<0.001$ and ****$P<0.0001$, unpaired Student's $t$ test). **h** Plot of $k_{off}$ for ATP-induced dissociation of BiP–substrate complex (as in **g**) against final concentration of MANF. The mean values and SD bars of three independent experiments are plotted. Single exponential best fit lines are shown. Source data are provided as a Source Data file

MANF's effects on BiP–nucleotide interactions. As Grp170 and MANF presumably bind on opposing sides of the NBD[19], this feature likely represents independent action by these two regulators of BiP activity (Supplementary Fig. 4c). MANF binding attenuated not only ADP release from BiP, but also the rate of binding of ATP to BiP in the nucleotide-free (apo) state (Fig. 5e, f). The functional consequences of MANF's ability to antagonize both aspects of nucleotide exchange were revealed by the observation that ATP-dependent release of a fluorescently labeled client peptide was retarded by the presence of MANF or its SAP domain (Fig. 5g, h).

Protein misfolding in the ER is associated with incorporation of BiP into detergent insoluble high-molecular-weight chaperone–client complexes[32] (Fig. 6a, compare lanes 1, 2, 5, and 6). In tunicamycin-treated $MANF^{-/-}$ CHO-K1 S21 cells, less BiP was recovered in such high-molecular-weight complexes when compared with wild-type cells (Fig. 6a, compare lanes 2, 4, 6, and 8); consistent with a role for MANF in stabilizing BiP in its substrate-bound ADP state.

To quantify this effect, we cultured the wild-type and $MANF$ knockout cells in isotopically labeled media, exposed the cells to tunicamycin to elicit protein misfolding in the ER, combined the lysates from the two genotypically divergent sources into a single sample, and analyzed the contribution of the two sources (wild-type and $MANF^{-/-}$) to the mass spectra of BiP peptides in the whole-cell extract and in the high-molecular-weight fraction of the combined sample (Fig. 6b). By eliminating the consequences of differences in sample loading, such stable isotope labeling with amino acids in cell culture (SILAC[33]) is rendered a powerful means to quantify differences in abundance of proteins from divergent sources that are processed experimentally as a single sample.

In a control experiment, we tested the effect of tunicamycin on wild-type cells (Supplementary Fig. 5a). As expected, tunicamycin led to an increase in the recovery of BiP in both the whole-cell extract and in the high-molecular-weight pellet (Supplementary Fig. 5b, c). Despite the fact that BiP's abundance in the whole-cell extract was similar in both genotypes, the contribution of BiP to the high-molecular-weight complexes from the $MANF$ knockout cells was lower than from the wild-type (Fig. 6c). Whilst the source of the variation in the extent of the genotype-based biased recovery of some BiP peptides over others remains unknown, the bias itself was a consistent finding, observed in SILAC experiments with heavy isotopes marking either genotype and over all 18 detectable BiP tryptic peptides (Fig. 6d). Furthermore, the suggestion that BiP–client interactions might be less stable in $MANF^{-/-}$ cells is also supported by the observation that about twofold less BiP was recovered in complex with the null Hong Kong variant of α1 antitrypsin (a model unfolded protein known

to interact with BiP[34]) in mutant versus wild-type cells (Supplementary Fig. 5d).

## Discussion

The findings presented here speak to an important intracellular role for MANF in maintenance of protein-folding homeostasis in the ER that may be independent of its activity as a secreted protein involved in intercellular communication. MANF's SAP domain engages BiP in the ADP-bound state and disfavors nucleotide exchange, thus acting as an NEI to modulate BiP's activity. The notion that MANF functions as an accessory factor regulating BiP activity fits well with the ubiquitous expression of the protein and its enrichment in secretory cells[35], and with the presence of a substantial pool of ER-localized MANF directed by the C-terminal ER retention motif[6]. BiP's essential role in protein-folding homeostasis in the ER readily explains both the basal activation of the UPR observed in $MANF$ knockout cells and the parallels between the phenotype of mice lacking MANF and mice lacking other components of the ER quality control network[7,36]. Recent findings, pointing to extensive internalization of extracellularly applied MANF[8], may reconcile evidence for a non-cell-autonomous component to the MANF-deficiency phenotypes with our findings on the basis for the intracellular actions of MANF.

The biophysical basis for MANF's NEI activity appears to be explained by stabilization of the ADP-bound or nucleotide-free (apo) conformation of the chaperone. Similar principles underlie the activity of mammalian Hip and yeast Sec72, the only other known Hsp70 NEIs[9,37], but the details vary in interesting ways. The TPR domains of Hip (and likely Sec72) form a bracket over the Hsp70's NBD, disfavoring the outward rotation of subdomain IIb and thereby inhibiting release of the bound nucleotide. TPR binding overlaps with the NEF binding site on the Hsp70 NBD, indicating that the Hip-bound chaperone would not be regulated by NEFs; though given the high dissociation rates of the Hip-Hsp70 complex, NEFs could likely compete favorably with Hip to stimulate nucleotide exchange. MANF contacts the opposite surface of the NBD; the cleft between subdomains Ia and IIa. Although nucleotide is released from the other end, MANF binding likely stabilizes BiP in the ADP-bound conformation, augmenting the energy barrier to nucleotide release from the BiP–MANF complex. As the MANF-bound NBD is free to interact with NEFs (Supplementary Fig. 4c), a putative ternary complex of MANF, BiP, and bound client would likely retain a measure of responsiveness to NEF; albeit attenuated.

The discovery of an ER-localized NEI adds an additional layer of complexity to the regulation of the Hsp70 chaperone BiP. At present, we have no information on how MANF integrates with the known ER-localized J-domain co-chaperones that stimulate

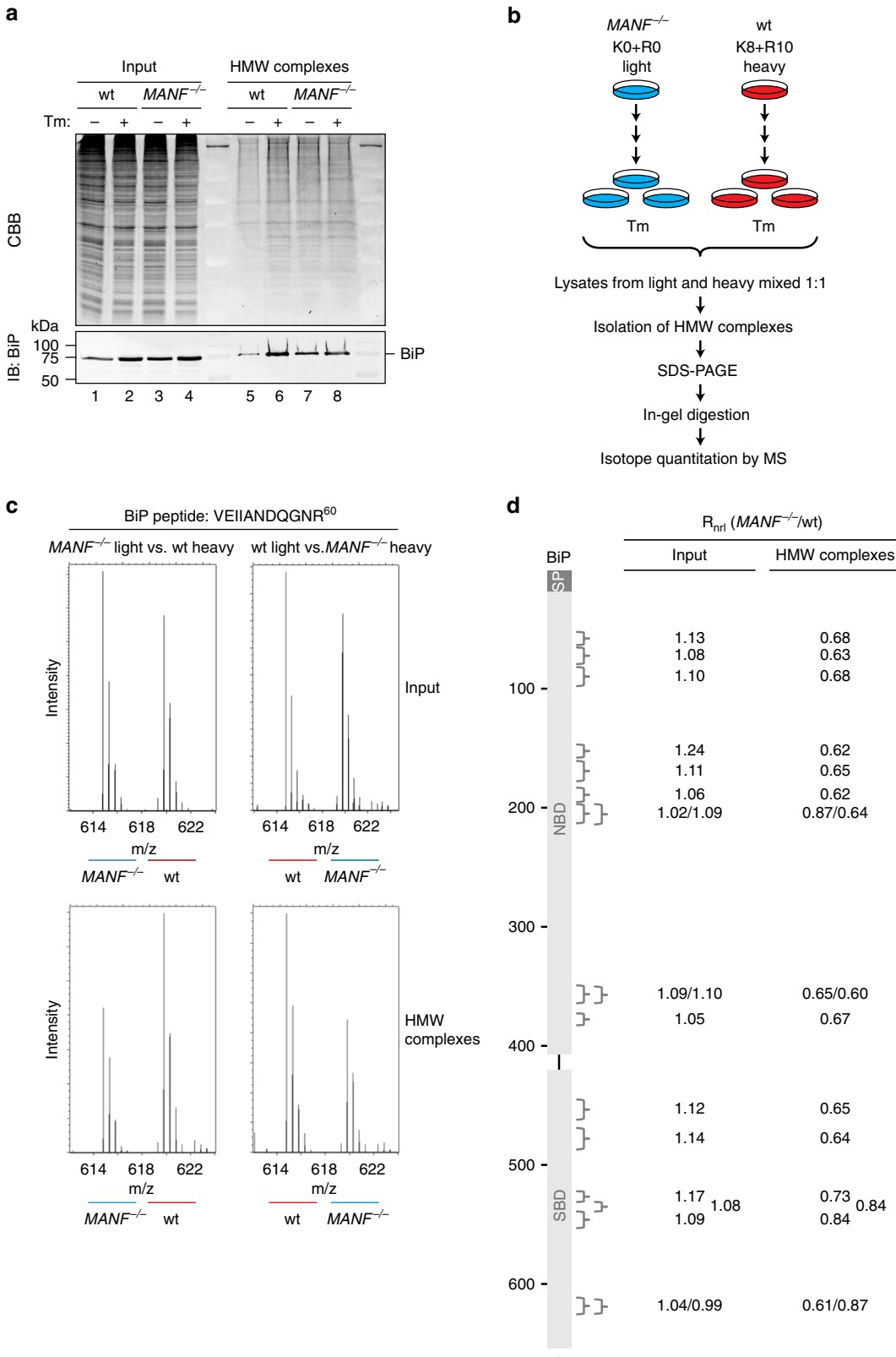

BiP association with its diverse clients[38] or with the two known ER-localized NEFs that promote the turnover of BiP–client complexes[39]. The relatively low affinity of the SAP domain for BiP's NBD suggests that the binary interaction we observe in vitro may reflect only part of the picture. Contacts between the SAPLIP domain and the NBD (observed crystallographically) do not

measurably contribute to the affinity of the binary interaction with BiP in solution, suggesting that they may be dispensable. Therefore, whilst contacts between SAPLIP, SAP, and the NBD trap the otherwise flexible SAPLIP-SAP interdomain linker in a particular conformation, in solution the SAPLIP domain may disengage from SAP and the NBD, as suggested by the NMR

**Fig. 6** MANF stabilizes high-molecular-weight BiP complexes in stressed cells. **a** Coomassie-stained (CBB) SDS-PAGE gel and corresponding BiP immunoblot of the cell lysate (Input) and detergent insoluble high-molecular-weight (HMW) complexes from untreated and tunicamycin-treated (Tm; 2.5 μg/mL, 15 h) CHO-K1 S21 wild-type (wt) and *MANF*-/- cells. Shown is a representative of three experiments. **b** Schema of the design of the SILAC experiment to quantify relative changes in abundance of BiP peptides incorporated into detergent-insoluble high-molecular-weight complexes in CHO-K1 S21 wild-type and *MANF*-/- cells treated with tunicamycin (Tm; 2.5 μg/mL, 15 h). **c** LC-MS spectra of a representative doubly charged tryptic BiP peptide (VEIIANDQGNR[60]) from the input (top) and HMW complexes (bottom) of experiments as outlined in (**b**). The spectrum on the left is from lysate of *MANF*-/- cells cultured in light medium combined with lysate of wild-type cells cultured in heavy medium, and the spectrum on the right is from lysate of *MANF*-/- cells cultured in heavy medium combined with lysate of wild-type cells cultured in light medium. **d** Averaged normalized ratios ($R_{nrl}$) of BiP peptides identified in the LC-MS spectra from the *MANF*-/- cells versus wild-type cells in the input and HMW complexes fraction from the two experiments as described in (**c**). The position of the peptides on the BiP sequence (654 amino acids) is indicated by the brackets. The BiP signal peptide (SP), nucleotide-binding domain (NBD), and substrate-binding domain (SBD) are indicated. Uncropped image for panel (**a**) and source data for panel (**d**) are provided as a Source Data file

studies[21,22]. It is thus tempting to consider that interactions between the mobile SAPLIP domain and either a subset of BiP clients or other co-regulators (for example J-proteins or NEFs), might direct MANF's NEI activity to specific BiP–client complexes, selectively stabilizing them. These speculative roles for the SAPLIP domain in the intracellular functions of MANF may co-exist with its recently recognized role as a sulfatide receptor, important for the non-cell-autonomous functions of MANF, including its uptake from the media into cells[8].

The parallels to Hip are intriguing. While the affinity of the binary Hip-Hsp70 NBD complex is low ($10^{-5}$ M; in the range of the NBD–MANF complex studied here), Hip is a multipartite protein with a client-binding domain that appears to contribute to its avidity to selective client-bound Hsp70 complexes (exemplified by the glucocorticoid receptor)[9,40]. Parallel ternary interactions in the ER (potentially involving the SAPLIP domain) could explain the recovery of BiP in complex with MANF. Furthermore, the enhancement of co-immunoprecipitation efficiency by chemical cross-linking is consistent with very high dissociation rates of the binary NBD–SAP complex observed in vitro.

The stabilization of BiP–client complexes may contribute to protein-folding homeostasis in and of itself, as the folding of some clients may require lengthier engagement with BiP than others. In the cytosol, Hip is believed to facilitate the handover of clients from Hsp70 to Hsp90, for maturation, or to CHIP, for destruction[9], and yeast Sec72 hands clients over from Ssb1 to the Sec61 translocon[37]. ER chaperones function in physically linked networks[41,42], to which MANF may connect, as possibly hinted by the recovery of both Grp94 and the NEF Grp170 in the MANF pull-down (Fig. 2a). It is thus tempting to speculate that MANF may stabilize certain BiP–client complexes so as to promote their transfer to specific downstream quality control effectors (Fig. 7). A hypothetical handover favored by the ability of MANF to function in a multi-component complex alongside NEFs would encourage context-specific nucleotide exchange and client release from BiP.

## Methods

**Cell lines**. All cells were grown on tissue culture dishes or multi-well plates (Corning) at 37 °C and 5% $CO_2$. CHO-K1 cells (ATCC CCL-61) were phenotypically validated as proline auxotrophs and their *Cricetulus griseus* origin was confirmed by genomic sequencing. *CHOP:GFP* and *XBP1s:Turquoise* reporters were introduced sequentially under G418 and puromycin selection to generate the previously described derivative CHO-K1 S21 clone[12]. The puromycin-resistance marker was subsequently lost, rendering CHO-K1 S21 cells sensitive to puromycin. The cells were cultured in Nutrient mixture F-12 Ham (Sigma) supplemented with 10% (v/v) serum (FetalClone II; HyClone), 1× Penicillin–Streptomycin (Sigma), and 2 mM L-glutamine (Sigma). HEK293T cells (ATCC CRL-3216) were cultured in DMEM (Sigma) supplemented as above. Cell lines were subjected to random testing for mycoplasma contamination using the MycoAlert Mycoplasma Detection Kit (Lonza).

Experiments were performed at cell densities of 70–90% confluence. Cells were treated with drugs at the following final concentrations: 0.5 μM thapsigargin

(Calbiochem) and 2.5 μg/ml tunicamycin (Melford) first diluted in fresh, pre-warmed medium, and then applied to the cells by medium exchange.

**Plasmid construction**. Supplementary Table 1 lists the plasmids used in this study. Standard PCR and molecular cloning methods were used to generate DNA constructs, and point mutations were introduced by PCR-based site-directed mutagenesis. Supplementary Table 2 lists all primers used in this study.

**MANF knockout using the CRISPR-Cas9 system**. Two single guide RNA sequences (plasmids UK1839 and UK1840) for targeting the third exon of *Cricetulus griseus* (Chinese hamster) *MANF* were selected from the CRISPy database [http://staff.biosustain.dtu.dk/laeb/crispy/[43]] and duplex DNA oligonucleotides of the sequences were inserted into the pSpCas9(BB)-2A-mCherry plasmid (plasmid UK1610) following published procedures[44]. In total, $2 \times 10^5$ CHO-K1 S21 cells were plated in six-well plates. Twenty-four hours later, the cells were transfected with 2 μg of guide RNA/Cas9 plasmids UK1839 and UK1840 using Lipofectamine LTX (Invitrogen). Thirty-six hours after transfection, the cells were washed with PBS, resuspended in PBS containing 4 mM EDTA and 0.5% (w/v) BSA, and mCherry-positive cells were individually sorted by fluorescence-activated cell sorting (FACS) into 96-well plates using a MoFlo Cell Sorter (Beckman Coulter). Clones were then analyzed by a PCR-based assay to detect *MANF* mutations as described previously[45]. Briefly, primers were designed for the region encompassing the *MANF* RNA guide target sites, and the reverse primer was labeled with 6-carboxyfluorescein (6-FAM) on the 5′ end. A PCR reaction was set up using 5 μl of AmpliTaq Gold 360 Master Mix (Applied Biosystems), 0.6 μl of a mix of forward and labeled reverse primers (each at 10 μM), 3.4 μl of $H_2O$, and 1 μl of genomic DNA (~10 ng/μl). PCR was performed as follows: 95 °C for 10 min, 10 × (94 °C for 15 s, 59 °C for 15 s, 72 °C for 30 s), 20 × (89 °C for 15 s, 59 °C for 15 s, 72 °C for 30 s), 72 °C for 20 min. PCR products were diluted 1:100 in water and fragment length was determined on a 3130xl Genetic Analyzer (Applied Biosystems), and the data were analyzed using the Gene Mapper software (Applied Biosystems). Clones for which frameshift-causing insertions or deletions were detected for both alleles were sequenced to confirm the *MANF* mutations.

**Retroviral production and transduction**. In an attempt to rescue the MANF-deficiency phenotype in CHO-K1 S21 *MANF*-/- cells, these cells were targeted with puromycin-resistant retrovirus expressing FLAG-M1-MANF. For that, HEK293T cells were split onto 6-cm dishes 24 h prior to co-transfection of pBABE Puro plasmids[46] empty or encoding FLAG-M1-MANF (UK2058 or UK2059, respectively) with VSV-G retroviral packaging vectors, using TransIT-293 Transfection Reagent (Mirus) according to the manufacturer's instructions. Sixteen hours after transfection, medium was changed to medium supplemented with 1% (w/v) BSA (Sigma). Retroviral infections were performed following a 24-h incubation by diluting 0.45-μm filter-sterilized cell culture supernatants at a 1:1 ratio into CHO-K1 S21 cell medium supplemented with 10 μg/ml polybrene (8 ml of final volume) and adding this preparation to CHO-K1 S21 *MANF*-/- cells ($1 \times 10^6$ cells seeded onto 10-cm dishes 24 h prior to infection). Infections proceeded for 8 h, after which the viral supernatant was replaced with fresh medium. Forty-eight hours later, the cells were split into 10-cm dishes containing medium supplemented with 6 μg/ml puromycin, and 24 h afterwards this was changed to medium supplemented with 8 μg/ml puromycin. The medium was changed every third day until puromycin-resistant colonies were visible. Three clones from each population were expanded and analyzed by flow cytometry (to assess UPR induction) and SDS-PAGE/immunoblotting (to check for FLAG-M1-MANF expression).

**Mammalian cell lysates**. Cell lysis was performed according to a previously published procedure[47] as follows: mammalian cells were cultured on 10-cm dishes and allowed to grow for ~36 h. Before lysis the dishes were placed on ice, washed with ice-cold PBS, and cells were detached in PBS containing 1 mM EDTA using a cell scraper. The cells were sedimented for 5 min at 370×g at 4 °C and lysed in lysis buffer [20 mM HEPES–KOH pH 7.4, 150 mM NaCl, 2 mM $MgCl_2$, 10% (v/v)

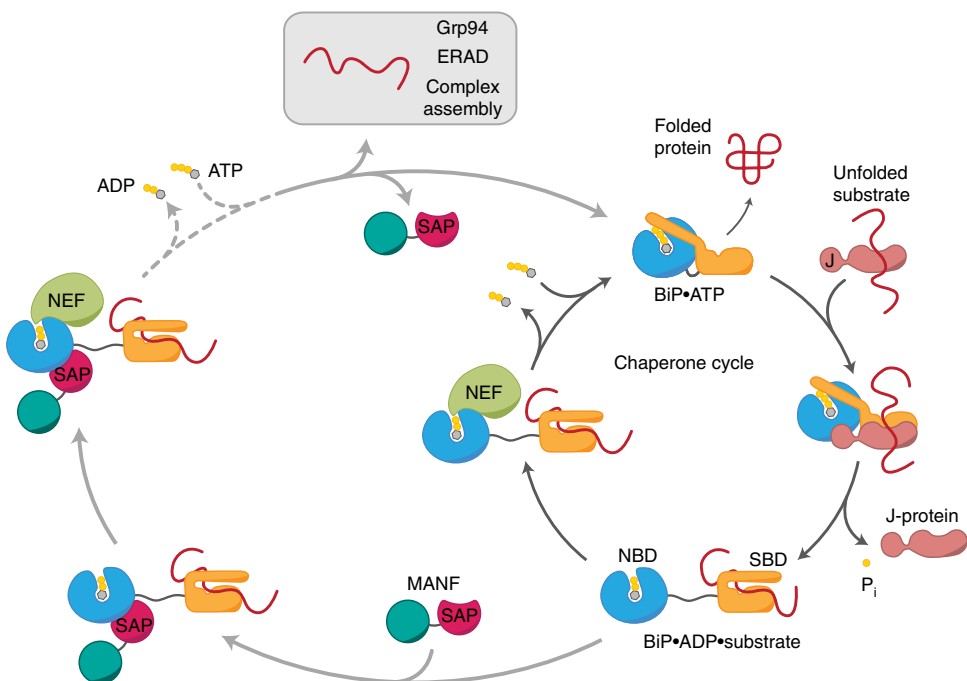

**Fig. 7** Cartoon depicting the function of MANF as a nucleotide exchange inhibitor (NEI) for BiP. ER-localized BiP undergoes a J-protein- and nucleotide exchange factor (NEF)-dependent chaperone cycle (dark gray). MANF can interact with substrate-bound BiP in the ADP state, in which the nucleotide-binding domain (NBD) and substrate-binding domain (SBD) are undocked, to slow down nucleotide exchange and substrate dissociation (dotted line). By analogy to the role of the cytosolic Hip protein, MANF-mediated stabilization of certain BiP–client complexes may enhance the efficiency of client transfer to downstream ER quality control effectors. These may include other chaperone systems (e.g., Grp94), degradation via ERAD, or factors involved in assembly of multimeric complexes. The binding of MANF via its SAP domain to the NBD of BiP allows the simultaneous action of NEFs and thereby adds an additional layer to the modulation of BiP's functional cycle

glycerol, 1% (v/v) Triton X-100] containing protease inhibitors (2 mM PMSF, 4 μg/ml pepstatin, 4 μg/ml leupeptin, 8 μg/ml aprotinin) for 10 min on ice. For FLAG-M1 immunoprecipitations cells were lysed in TBS/Ca$^{2+}$ lysis buffer [50 mM Tris-HCl pH 7.4, 150 mM NaCl, 10 mM CaCl$_2$, 10% (v/v) glycerol, 1% (v/v) Triton X-100] containing protease inhibitors. The lysates were cleared for 10 min at 21,000× $g$ at 4 °C. Bio-Rad protein assay reagent was used to determine the protein concentrations of lysates followed by normalization. For analysis by SDS-PAGE, SDS sample buffer was added to the lysates and proteins were denatured by heating for 10 min at 70 °C before separation on 12.5% SDS polyacrylamide gels.

**Immunoblot analysis**. After separation by SDS-PAGE, the proteins were transferred onto PVDF membranes. Membranes were blocked with 5% (w/v) dried skimmed milk in TBS (25 mM Tris-HCl pH 7.5, 150 mM NaCl) and incubated with primary antibodies followed by IRDye fluorescently labelled secondary antibodies (LI-COR). Membranes were scanned with an Odyssey near-infrared imager (LI-COR). Primary antibodies against hamster BiP (chicken anti-BiP, 1:2000;[48]), actin (mouse anti-actin, 1:2000; Abcam, cat. # AB3280), MANF (chicken anti-MANF, 1:1000; see below), FLAG-M1 (mouse anti-FLAG-M1, 1:1000; Sigma, cat. # F3040), and A1AT (mouse anti-A1AT monoclonal, 1:5000; Abcam, cat. # AB9399) were used.

**Flow cytometry**. The effect of MANF-deficiency on UPR signaling was analyzed by flow cytometry. The sensitivity to UPR induction was tested by treating cells with the UPR-inducing compound, thapsigargin, for 16 h before analysis. Briefly, cells were washed with PBS and collected in PBS containing 4 mM EDTA, and single-cell fluorescence signals (20,000/sample) were analyzed by dual-channel flow cytometry with an LSRFortessa cell analyzer (BD Biosciences). GFP and mCherry fluorescence was detected with excitation laser 488 nm, filter 530/30, and excitation laser 561, filter 610/20, respectively. Data were processed using FlowJo and median reporter analysis was performed using Prism 5 (GraphPad). An example of the gating strategy is presented in Supplementary Fig. 6.

**Immunoprecipitation from cell culture supernatants**. In order to assess the relative amount of MANF that is secreted (Fig. 1d), CHO-K1 S21 *MANF*$^{-/-}$ cells and *MANF*$^{-/-}$ cells stably expressing FLAG-M1-MANF were grown in six-well plates for 48 h, and at time = 0 the medium was changed to 1 ml of medium without serum supplemented with thapsigargin. At the indicated times, the cell

culture supernatants were collected, centrifuged for 10 min at 3000× $g$ at 4 °C, and 900 μl were transferred to a new tube containing 100 μl of 10 × TBS/Ca$^{2+}$ lysis buffer without glycerol. For immunoprecipitation of FLAG-M1-MANF, 15 μl ANTI-FLAG-M1 Agarose Affinity Gel (Sigma, cat. # A4596) were washed in TBS/Ca$^{2+}$ lysis buffer without glycerol and then added to the prepared cell culture supernatants, and incubated overnight at 4 °C. The beads were then recovered by centrifugation for 1 min at 845× $g$ and washed three times for 3 min at 4 °C with TBS/low Ca$^{2+}$ buffer [50 mM Tris-HCl pH 7.4, 150 mM NaCl, 2 mM CaCl$_2$, 0.1% (v/v) Triton X-100] containing protease inhibitors. Bound proteins were eluted by addition of 2 × SDS sample buffer and heating for 10 min at 70 °C. Equal volumes of the samples were loaded on a 12.5% SDS polyacrylamide gel. Cells from the same dishes were lysed in TBS/Ca$^{2+}$ lysis buffer, and samples of the normalized cell lysates (45 μg) were loaded as a control. FLAG-M1-MANF, BiP, and actin were detected by immunoblotting.

In order to assess the efficiency of the FLAG-M1 immunoprecipitation and FLAG-M1-MANF stability in the cell culture medium (Supplementary Fig. 1c), 3, 15, and 30 μg of total protein extracted from CHO-K1 S21 *MANF*$^{-/-}$ cells stably expressing FLAG-M1-MANF were added to serum-free medium and incubated for 0 or 120 min under the same conditions as the ones where cells are grown. At the end of the incubation time, a FLAG-M1 immunoprecipitation was performed and bound proteins were eluted as above. Equal volumes of the immunoprecipitation samples and samples of the cell lysates were loaded as an "input" control on a 12.5% SDS polyacrylamide gel, and analyzed by immunoblotting.

**In vivo cross-linking and co-immunoprecipitation**. In order to confirm BiP–MANF interaction in vivo, CHO-K1 S21 *MANF*$^{-/-}$ cells and *MANF*$^{-/-}$ cells stably expressing FLAG-M1-MANF were subjected to in vivo cross-linking, followed by FLAG-M1 co-immunoprecipitation and analysis by mass spectrometry. In vivo cross-linking was performed following a previously published protocol[49] with modifications. Cells were grown in 10-cm dishes (six plates per sample) and washed twice with PBS containing 0.1 mM CaCl$_2$ and 1 mM MgCl$_2$. For cross-linking, 1 mM dithiobis(succinimidyl propionate) (DSP; Thermo Scientific Pierce, cat. # 22585) was prepared by dilution into pre-warmed PBS (37 °C) containing 0.1 mM CaCl$_2$ and 1 mM MgCl$_2$, and exposed to the cells for 2 h on ice. Then, the DSP-containing solution was removed and the residual DSP was quenched by incubating the cells for 15 min with PBS containing 0.1 mM CaCl$_2$, 1 mM MgCl$_2$, and 2 mM Tris-HCl pH 7.4. After removing the quenching solution, the cells were

washed with PBS containing 0.1 mM $CaCl_2$ and 1 mM $MgCl_2$, and then detached in PBS using a cell scraper. The cells were sedimented for 5 min at $370\times g$ at 4 °C and lysed in TBS/$Ca^{2+}$ lysis buffer for 10 min on ice. A post-nuclear supernatant was prepared by centrifugation for 5 min at $800\times g$ at 4 °C, and then transferred into a new reaction tube and cleared twice for 10 min at $21,000\times g$ at 4 °C. Bio-Rad protein assay reagent was used to determine the protein concentrations of lysates followed by normalization. For immunoprecipitation of FLAG-M1-MANF, 40 µl ANTI-FLAG-M1 Agarose Affinity Gel were washed in TBS/$Ca^{2+}$ lysis buffer without glycerol and then added to the cleared lysates and incubated 2 h at 4 °C. The beads were then recovered by centrifugation for 1 min at $845\times g$ and washed three times at 4 °C with TBS/low $Ca^{2+}$ buffer, containing protease inhibitors. Bound proteins were eluted by addition of $2 \times$ SDS sample buffer and heating for 10 min at 70 °C. Equal volumes of the samples were loaded on a 12.5% SDS polyacrylamide gel. Samples of the normalized cell lysates (15 µg) were loaded as an "input" control. The gel was stained with Coomassie (InstantBlue; Expedeon), and selected bands were cut out for in-gel digest with trypsin endopeptidase and analysis by mass spectrometry.

The interaction of endogenous BiP with the FLAG-tagged null Hong Kong variant of α1-antitrypsin (AAT-NHK-QQQ-3×FLAG) was analyzed according to a previously described procedure[34] as follows: CHO-K1 S21 wild-type and $MANF^{-/-}$ cells were mock transfected or transfected with 4 µg of plasmid UK2283 using Lipofectamine LTX (Supplementary Fig. 5d). Forty hours after transfection, cells were harvested as described above. The lysates were cleared twice, normalized for their protein content, and equal quantities of the lysates were incubated with 15 µl of ANTI-FLAG-M2 beads (Sigma, cat. # A2220) for 1 h at 4 °C. The beads were then recovered by centrifugation for 1 min at $845\times g$ and washed three times for 3 min at 4 °C with lysis buffer containing protease inhibitors. Bound proteins were eluted by addition of $2 \times$ SDS sample buffer and heating for 10 min at 70 °C. Equal volumes of the samples were loaded on a 12.5% SDS polyacrylamide gel. Samples of the normalized cell lysates (60 µg) were loaded as a control. AAT-NHK-3×FLAG and endogenous BiP were detected by immunoblotting with FLAG-M2- and BiP-specific antibodies.

**Isolation of high-molecular-weight complexes.** Isolation of high-molecular-weight complexes was performed following a modified protocol described previously[32]. Cells cultured in 10-cm dishes (four plates per sample) were grown for 24 h, then the medium was exchanged before treatment with tunicamycin for 15 h. Cells were then washed with ice-cold PBS and detached in PBS containing 2 mM EDTA and 20 mM N-ethylmaleimide (NEM) using a cell scraper. The cells were sedimented for 5 min at $370\times g$ at 4 °C and lysed in 120–200 µl of buffer BP [20 mM HEPES–KOH pH 7.5, 250 mM sucrose, 100 mM NaCl, 2.5 mM $CaCl_2$, 20 mM NEM, 10% (v/v) glycerol, 1% (v/v) Triton X-100] containing protease inhibitors, for 5 min on ice. A post-nuclear supernatant was prepared by centrifuging for 5 min at $800\times g$ at 4 °C. Bio-Rad protein assay reagent was used to determine the protein concentrations of lysates followed by normalization. Equal amounts of total protein (1–1.5 mg) were brought to room temperature and adjusted to 1% (v/v) SDS in a final volume of 150 µl. This sample was then layered on a $2 \times$ volume (300 µl) cushion [20% (v/v) glycerol, 20 mM HEPES–KOH pH 7.4, 0.5% (v/v) Triton X-100, 0.8% (v/v) SDS], followed by ultra-centrifugation for 45 min at $100,000\times g$ at 4 °C (using a TLA-110 rotor; Beckman Coulter). The supernatant was then carefully removed without disturbing the pellet. The pellet was resuspended in 25 µl of urea loading buffer [8 M urea, 40 mM Tris-HCl pH 6.8, 1.36% (v/v) SDS, 0.002% (w/v) bromophenol blue, 7.5% (v/v) glycerol, 100 mM DTT], by shaking 5 min in a horizontal shaker at 700 rpm at 23 °C. After 1 h further incubation at 23 °C, the resuspended pellet was heated at 30 °C for 5 min before loading on a 12.5% SDS polyacrylamide gel alongside samples of the normalized cell lysates (30 µg) used as an "input" control. The gels were stained with Coomassie and analyzed by immunoblotting.

**Preparation of SILAC samples.** The experimental strategy for the stable isotope labeling by amino acids in cell culture (SILAC) experiment[33] is outlined in Fig. 6b, and samples were prepared as follows: CHO-K1 S21 wild-type and $MANF^{-/-}$ cells were adapted to Ham's F12 medium minus L-arginine and L-lysine for SILAC (Pierce, cat. # 88424) supplemented with 10% (v/v) dialyzed fetal bovine serum (Gibco, cat. # 26400-044), 1 x Penicillin–Streptomycin (Sigma), 2 mM L-glutamine (Sigma), 280 mg/l L-proline (Sigma, cat. # P5607), 62.5 mg/l L-lysine mono-hydrochloride (referred to as light lysine; Sigma, cat. # L8662) and 60.5 mg/l L-arginine monohydrochloride (referred to as light arginine; Sigma, cat. # A6969), and incubated as described above. Once adapted, the cells were cultured in SILAC medium containing either light L-lysine monohydrochloride and L-arginine monohydrochloride (as above) or R10 L-arginine monohydrochloride (referred to as heavy arginine; CK Isotopes, cat. # CNLM-539) and K8 L-lysine monohydrochloride (referred to as heavy lysine; CK Isotopes, cat. # CNLM-291) for several passages (>15 cell divisions) before expansion in four 10-cm dishes per sample. The cells were grown to 70–90% confluence, and the medium with or without tunicamycin was exchanged 15 h before harvesting, as described above (Isolation of high-molecular-weight complexes). After normalization of the post-nuclear supernatants for equal total protein concentration (by the Bio-Rad protein assay reagent), equal volumes of each sample were mixed (1:1 ratio) as indicated

and high-molecular-weight complexes were isolated as described above. Samples of the normalized cell lysates (30 µg) were used as an "input" control.

**Mass spectrometry.** The SILAC samples were resolved ~2 cm into a pre-cast 4–12% Bis-Tris polyacrylamide gel. The lanes were excised and cut in four equal pieces, and the proteins were reduced, alkylated, and digested in-gel with trypsin endopeptidase. The resulting tryptic peptides were analyzed by LC-MS/MS using a Q Exactive Plus coupled to a RSLCnano3000 (Thermo Scientific). Peptides were resolved on a 50-cm EASY-spray column (Thermo Scientific) using a gradient rising from 10 to 40% solvent B (80% MeCN, 0.1% formic acid) by 42 min. MS spectra were acquired at 70,000 (fwhm) between m/z 400 to 1500. MSMS data were acquired in a 10 top DDA fashion. Data were processed using Maxquant 1.5.8.3 with searches performed against a UniProt Chinese hamster and CHO database (downloaded 09/05/2018 and 10/05/2018 with 34,717 and 23,884 entries, respectively). Carbamidomethyl (C) was set as a fixed modification with oxidation (M) and acetyl (protein N-terminus) as variable modifications. Protein and peptide FDR were set at 1% and the re-quant function was turned off.

**Protein expression and purification.** Full-length mouse MANF (wild-type and mutants; UK2006, UK2209, UK2210, UK2212, UK2280) and SAP domain (126-169; UK2079) were expressed in the Origami B (DE3) E. coli strain (New England BioLabs, cat. # C3029). These constructs all contained an N-terminal His$_6$-Smt3 tag. The bacterial cultures were grown at 37 °C to an optical density ($OD_{600}$) of 0.8 in $2 \times$ TY medium supplemented with 100 µg/ml ampicillin, and expression of recombinant protein was induced at 22 °C for 16 h by the addition of 0.5 mM isopropylthio β-D-1-galactopyranoside (IPTG). After harvesting by centrifugation, the pellets were suspended in the HisTrap column-binding buffer (20 mM Tris-HCl pH 7.4, 0.5 M NaCl, 20 mM imidazole) containing Benzonase nuclease (1000 U per 1 l of expression culture; Sigma). The cells were crushed by a cell disruptor (Constant systems) at 30 kPSI and the obtained lysates were cleared by centrifugation for 1 h at $45,000\times g$. The supernatant was applied to a pre-equilibrated 5 ml HisTrap column (GE Healthcare) using a peristaltic pump. After washing with about 100 ml of the binding buffer, the bound fusion protein was eluted with 20–200 mM imidazole gradient using a FPLC purifier system (ÄKTA; GE Healthcare). Peak fractions were pooled and digested with SENP2 protease (at final 0.01 mg/ml; produced in-house) to cleave off the His$_6$-Smt3 tag overnight at 4 °C. After complete digestion, the remaining full-length fusion protein and His$_6$-Smt3 tag were removed by binding back to a HisTrap column. Following buffer exchange to lower salt buffer (10 mM HEPES–KOH pH 7.4, 50 mM NaCl), the intact protein was further purified by cation exchange chromatography using a HiTrap SP HP column (GE Healthcare) and eluted by the 50–500 mM NaCl gradient in 10 mM HEPES–KOH pH 7.4. The elution peak fractions were concentrated using centrifugal filters (Amicon Ultra, 10 kDa MWCO; Merck Millipore) and proteins were snap-frozen in liquid nitrogen and stored at −80 °C. Samples of fractions from each purification step were analyzed by SDS-PAGE and Coomassie staining. Protein samples for crystallization were further purified using a HiLoad 16/600 Superdex 75 prep grade gel filtration column equilibrated with 10 mM Tris-HCl pH 7.4 and 0.15 M NaCl.

N-terminal His$_6$-Smt3 Chinese hamster BiP NBD (28–413; UK2039, UK2022) was expressed in M15 E. coli cells (Qiagen) (as described above in $2 \times$ TY containing 100 µg/ml ampicillin and 50 µg/ml kanamycin) and purified by sequential nickel chelating, His$_6$-Smt3 tag cleavage, reverse nickel chelating, anion exchange chromatography using a HiTrap Q HP column (GE Healthcare), and size-exclusion chromatography, as described above. The lidless Chinese hamster BiP$^{V461F}$ (28–549; UK2121) was expressed in M15 E. coli cells and purified by sequential nickel chelating, anion exchange and size-exclusion chromatography.

N-terminally GST-tagged MANF or individual MANF domains (UK2013, UK1987, UK2004, UK2005) were used for biochemical assays. Protein expression was performed in Origami B (DE3) E. coli cells and induced as described above. The bacterial pellets were re-suspended in PBS containing Benzonase nuclease. The supernatants of the cell lysates were applied to a 5 ml GSTrap 4B prepacked column (GE Healthcare) using a peristaltic pump. After washing with 100 ml of PBS, the fusion protein was eluted with 30 mM Tris-HCl pH 7.4, 0.1 M NaCl, and 40 mM reduced glutathione. Fractions containing the fusion protein were collected and digested with TEV protease at 100:1 (protein:protease) molar ratio for 16 h at 4 °C. The GST-tag was removed by binding back to a GSTrap 4B column.

N-terminal His$_6$-Smt3 human Grp170 (UK2225) was expressed in E. coli BL21 T7 Express $lysY/I^q$ cells (New England BioLabs, cat. # C3013) and induced and purified by sequential nickel chelating, His$_6$-Smt3 tag cleavage, reverse nickel chelating, and anion exchange chromatography, as described above.

Full-length wild-type and T229A-V461F mutant Chinese hamster BiP proteins carrying an N-terminal His$_6$ tag (UK173 and UK1825) were expressed in M15 E. coli cells. The bacterial cultures were grown at 37 °C to $OD_{600}$ 0.8 in LB medium containing 50 µg/ml kanamycin and 100 µg/ml ampicillin. Protein expression was induced with 1 mM IPTG and cells were further incubated at 37 °C for 6 h. The cells were harvested by centrifugation and lysed with a high-pressure homogenizer (EmulsiFlex-C3; Avestin) in buffer A [50 mM Tris-HCl pH 7.5, 500 mM NaCl, 1 mM $MgCl_2$, 0.2% (v/v) Triton X-100, 10% (v/v) glycerol, 20 mM imidazole] containing protease inhibitors and 0.1 mg/ml DNaseI. The lysates were centrifuged

for 30 min at 25,000× $g$ and incubated with 1 ml Ni-NTA agarose (Qiagen) per 1 l of expression culture for 2 h rotating at 4 °C. The matrix was then transferred to a gravity-flow column and washed with buffer B [50 mM Tris-HCl pH 7.5, 500 mM NaCl, 0.2% (v/v) Triton X-100, 10% (v/v) glycerol, 30 mM imidazole] followed by buffer C [50 mM HEPES–KOH pH 7.4, 300 mM NaCl, 5% (v/v) glycerol, 10 mM imidazole, 5 mM β-mercaptoethanol] and further wash steps in buffer C supplemented sequentially with (i) 1% (v/v) Triton X-100, (ii) 1 M NaCl, (iii) 3 mM Mg$^{2+}$-ATP, or (iv) 0.5 M Tris-HCl pH 7.5. After a further wash step in buffer C containing 35 mM imidazole, the BiP proteins were eluted with buffer D [50 mM HEPES–KOH pH 7.5, 300 mM NaCl, 5% (v/v) glycerol, 5 mM β-mercaptoethanol, 250 mM imidazole] and dialyzed against HKM buffer (50 mM HEPES–KOH pH 7.4, 150 mM KCl, 10 mM MgCl$_2$). The proteins were concentrated, snap-frozen in liquid nitrogen, and stored at −80 °C.

Chicken anti-mouse MANF immunoglobulin was produced by Gallus Immunotech (Ontario, Canada) using bacterially expressed full-length MANF (UK 2006) produced as described above.

**Bio-layer interferometry**. In vitro biotinylation of proteins for use as ligands in Bio-layer interferometry (BLI) experiments was performed according to a previously published protocol[50] as follows: N-terminal fusion proteins of AviTag to MANF$^{22-179}$ (UK1987), SAPLIP$^{22-123}$ (UK2004), SAP$^{119-179}$ (UK2005), and BiP$^{19-413}$ (NBD, UK2022) were expressed in E. coli and purified as described above. After cleavage of the purification tag (GST-TEV or His$_6$-Smt3, to expose the N-terminal AviTag), the purified protein was adjusted to a final concentration of 20–50 μM in biotinylation buffer (25 mM Tris-HCl pH 7.4, 50 mM NaCl, 0.3 mM TCEP, 0.2 mM biotin, 2 mM MgCl$_2$, 2 mM ATP) and biotinylation was initiated by adding purified E. coli BirA (produced in-house) at a final concentration of 0.1–0.5 μM. The reaction was allowed to progress at 30 °C for 0.5-2 h. Excess biotin was removed by buffer exchange into 50 mM HEPES–KOH pH 7.4 and 100 mM KCl. Biotinylation was judged as complete by monitoring the fraction of AviTagged protein that was shifted in mobility during SDS-PAGE by an excess of Streptavidin[50] and the protein was stored frozen (−80 °C) in small aliquots until use.

Experiments were performed on an Octet RED96 (Pall ForteBio) in HKMT buffer [50 mM HEPES–KOH pH 7.4, 100 mM KCl, 10 mM MgCl$_2$, 0.05% (v/v) Triton X-100]. Streptavidin biosensors were loaded with biotinylated components as indicated to ~2 nm, washed in buffer, and then dipped into wells containing analytes to record association, and finally dipped into buffer well for dissociation. The normalized baseline, association, and dissociation signals are shown in Fig. 2c. In the sequential dipping experiment (Fig. 2d and Supplementary Fig. 2a), streptavidin biosensors were loaded with biotinylated MANF or SAP or SAPLIP, washed in buffer, and then sequentially dipped in wells containing BiP SBD, buffer, BiP NBD with 2 mM ATP, 2 mM ATP, BiP NBD with 2 mM ADP and 2 mM ADP. The normalized signals after the first wash step are shown. In the steady-state analysis of BiP NBD and MANF/SAP-binding affinity (Fig. 2e, Supplementary Fig. 2b, c, and Supplementary Fig. 4d), streptavidin biosensors were loaded with biotinylated NBD and dipped into wells containing various concentrations of MANF or SAP. Signals at equilibrium were plotted against analyte concentrations. $K_{1/2\ max}$ values were calculated by fitting data from three independent experiments to a one-site-specific binding function using Prism 6.

**Single-turnover ATPase assay**. Single-turnover experiments were performed according to a previously described procedure[51] as follows: complexes of BiP with ATP were formed in 50-μl reactions containing 30 μM wild-type BiP protein (UK173), 800 μM ATP, 0.444 MBq [α-$^{32}$P]-ATP (EasyTide; Perkin Elmer) in 25 mM HEPES–KOH pH 7.4, 100 mM KCl, 10 mM MgCl$_2$ for 3 min on ice, and isolated by gel filtration using a illustra Sephadex G-50 NICK column (GE Healthcare) pre-saturated with 1 mg/ml BSA solution and equilibrated in reaction buffer. The complexes were flash-frozen in aliquots and stored at −20 °C. For each reaction, a fresh aliquot was thawed, and after a zero time-point sample has been withdrawn, the remaining complexes were added to reactions containing MANF proteins at the indicated concentrations in the same buffer. The reactions were incubated at 30 °C and samples were taken at the indicated time points and directly spotted onto a thin-layer chromatography (TLC) plate. At the end of the time-course, the nucleotides were separated by developing the TLC plate with 400 mM LiCl and 10% (v/v) acetic acid as a mobile phase. The radioactive signals were detected by autoradiography and quantified. As a positive control, a reaction was performed containing the J-domain of ERdj6 fused to GST (J; UK185).

**Crystallization**. BiP NBD (UK2039) was mixed with an equimolar concentration of MANF (UK2006) or SAP (UK2079) before setting up 96-well screening trays. For the NBD–SAP complex, initial multiple needle crystals were obtained with wizard classic screen 4 (Emerald Biosystem) at 20 °C. These initial crystals grown in 2.4 M sodium malonate and 2.1 mM DL-malic acid pH 7.0 were used as seeds for micro-seeding. Final crystals for data collection grew at 1.5 mM (36 mg/ml) protein concentration in a 96-well sitting-drop plate with 200 nl protein solution, 150 nl well solution (1.92 M sodium malonate), and 50 nl diluted seeds. Crystals were briefly soaked into 1.9 M sodium malonate containing 30% (v/v) glycerol and cryocooled in liquid nitrogen.

Initial NBD–MANF complex crystals were obtained with 12% (w/v) PEG 6000, 0.1 M Tris-HCl pH 7.5 from a ProPlex screen (Molecular Dimensions) at 20 °C and used as seeds for micro-seeding. Final crystals for data collection grew at 1.0 mM (30 mg/ml) protein concentration in a 96-well sitting-drop plate set up with 200 nl protein solution, 150 nl well solution [7% (w/v) PEG 6000, 0.1 M Tris-HCl pH 7.5], and 50 nl diluted seeds. Crystals were briefly soaked into 7% (w/v) PEG 6000, 0.1 M Tris-HCl pH 7.5 containing 25% (v/v) ethylene glycol and cryocooled in liquid nitrogen.

Lidless BiP$^{V461F}$ (UK2121) was used to co-crystallize with MANF or SAP. Many hits were obtained with a PEG/ion screen (Hampton research), but only BiP$^{V461F}$ was present in the solved crystal structure. Final crystals for data collection grew in 8% (w/v) PEG 1000, 0.1 M Tris-HCl pH 8.5, and cryoprotected with extra 25% (v/v) ethylene glycol.

**Data collection and structure determination**. Diffraction data were collected at beamlines in the Diamond Synchrotron Light Source (DLS) (Supplementary Table 3) and processed by the XIA2 pipeline[52] implementing Mosflm[53] or XDS[54] for indexing and integration, Pointless[55] for space group determination, and Aimless[55] for scaling and merging. All three structures were solved by molecular replacement using Phaser[56]. For solving the NBD–SAP complex structure, the NBD moiety was searched using PDB 3LDN as a template and the SAP moiety was built manually in COOT[57] based on the high-resolution electron density map. Then this NBD–SAP complex and SAPLIP domain from PDB 2W51 were used as search models for solving the structure of NBD–MANF complex. The linker of MANF was built manually according to the electron density in COOT. The structure of BiP$^{V461F}$ (apo) was solved by searching the NBD (PDB 3LDN) and SBD (PDB 5E85) as models, respectively. Further refinement was performed iteratively using COOT, phenix.refine[58], and refmac5[59] (Supplementary Table 3). MolProbity[60] was consulted throughout the refinement process, at the end of which 99.5% of residues from the two complexes and 97.5% from BiP$^{V461F}$ were in the favored Ramachandran region and none were outliers. Molecular graphics were generated using UCSF Chimera[61] and PyMol (The PyMOL Molecular Graphics System, Version 1.3 Schrödinger, LLC).

**Fluorescence measurement**. Nucleotide release kinetics of the fluorescent nucleotide analogue MABA-ADP (Jena Bioscience, cat. # NU-893-MNT) and nucleotide binding of MABA-ATP (Jena Bioscience, cat. # NU-806-MNT) were performed in a Perkin Elmer LS55 fluorescence spectrometer instrument (excitation 360 nm, emission 420 nm). These nucleotide analogues carry a fluorescent MANT group attached to the ribose, which shows a significant increase of fluorescence upon Hsp70 binding[31].

MABA-ADP release assay: BiP or NBD (UK173, UK2039) and MABA-ADP complexes were formed by mixing 2.5 μM BiP or NBD and an equimolar concentration of MABA-ADP, and incubating at 30 °C for 3 h. The nucleotide exchange solutions contained 250 μM ATP or ADP and protein factors as indicated. The reaction buffer used was 50 mM HEPES–KOH pH 7.4, 100 mM KCl, and 10 mM MgCl$_2$. Equal volumes of the solutions were mixed in a 50 μl quartz cuvette (Hellma Analytics) and the release of MABA-ADP was monitored by the decrease in MABA-ADP fluorescence. The curves were fitted to a one phase exponential decay function (Prism 5) to calculate the release rate of MABA-ADP. Rates were determined from three independent experiments. Stimulated MABA-ADP release experiments (Fig. 5d) were performed likewise but in presence of 0.5 mM CaCl$_2$ and by mixing pre-formed complexes of BiP and MABA-ADP 1:1 with a solution containing 2.5 μM Grp170 (UK2225) and 250 μM ATP.

MABA-ATP-binding assay: 2.5 μM BiP (UK173) was mixed with an equal volume of 2.5 μM MABA-ATP in the presence of 200 μM MANF (UK2006) or its mutants (UK2013, UK2079, UK2209, UK2210) in a 50-μl quartz cuvette. The binding of MABA-ATP was monitored by measuring the increase of MABA-ATP fluorescence. The curves were fitted to a one-phase exponential association model (Prism 5) to calculate the observed binding rate of MABA-ATP. Rates were determined from three independent experiments.

**Fluorescence polarization measurements**. Purified BiP protein (UK173) at 40 μM was incubated with 1.5 μM lucifer yellow (LY)-labeled substrate peptide (HTFPAVLGSC;[62]) in the presence of 0.5 mM ADP overnight at room temperature in FP buffer [50 mM HEPES–KOH pH 7.4, 100 mM KCl, 10 mM MgCl$_2$, 1 mM CaCl$_2$, 0.01% (v/v) Triton X-100]. Twenty microliters of pre-formed BiP–substrate complexes were transferred to a 386-well polystyrene microplate (μClear, black; Greiner Bio-One, cat. # 781096) and mixed with 5 μl of 20 mM ATP (final 4 mM), and various concentrations of purified wild-type and mutant MANF proteins (UK2006, UK2013, UK2079, UK2209, and UK2210) in FP buffer. Fluorescence polarization of the lucifer yellow fluorophore was measured (excitation 430 nm; emission 535 nm) at room temperature in a plate reader (Infinite F500; Tecan). The curves were fitted to a one-phase exponential decay (Prism 5) to calculate the rate of ATP-induced substrate peptide release from BiP in the presence of MANF or its mutants. Rates were determined from three independent experiments.

## Data availability

Atomic coordinates of the X-ray structures were deposited in the Protein Data Bank (PDB) with accession codes 6H9U (BiP NBD in complex with MANF SAP), 6HA7 (BiP NBD in complex with MANF), and 6HAB (BiP$^{V461F}$ in the apo state). The source data underlying Figs. 1c, 4d, 5a–h, 6d and Supplementary Figs 1b, 4a, b, and 5c, d are provided as a Source Data file. Other data are available from the corresponding authors upon reasonable request.

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

## Acknowledgements

We thank colleagues from the CIMR, University of Cambridge: Luke A. Perera for advice and useful discussions, Randy Read for advice on analyzing the crystallographic data, Robin Antrobus for mass spectrometry analyses, the CIMR flow cytometry core facility team (Reiner Schulte, Chiara Cossetti, and Gabriela Grondys-Kotarba) for assistance with FACS, the Huntington lab for access to the BLI Octet machine, and Niko Amin-Wetzel and Ana Crespillo-Casado for comments on the paper. The Grp170 expression plasmid was a kind gift from Claes Andréasson (Stockholm University). Supported by Wellcome Trust Principal Research Fellowships to DR (Wellcome 200848/Z/16/Z) and a Wellcome Trust Strategic Award to the Cambridge Institute for Medical Research (Wellcome 100140).

## Author contributions

D.R., C.R., and S.P. conceived the project. D.R., S.P., C.R., and Y.Y. designed experiments. C.R. carried out cell culture experiments. S.P., L.R., and D.R. performed initial biochemical characterizations. Y.Y. crystallized proteins, solved the structures, and performed biochemical experiments. Y.Y., C.R., S.P., D.R., and L.R. analyzed the data. D.R. wrote the initial draft of the paper. Y.Y., C.R., and S.P. prepared the figures and contributed to writing the paper. D.R. acquired funding and supervised the project.

## Additional information

**Competing interests:** The authors declare no competing interests.

