## [Peer Review File · Nature Communications]

Reviewers' comments:

Reviewer #1 (Remarks to the Author):

The manuscript by Yan et al. reports a novel function of MANF as a NEI for BiP and determined co-crystal structures of MANF in complex with BiP. Although MANF was initially characterized as a secreted protein involved in intercellular communication, a number of studies suggested MANF is also present in ER, plays a role in ER protein homeostasis, and associates with BiP. Based on these observations, the author first confirmed MANF interacts with BiP, and demonstrated a physical interaction between MANF and the NBD of BiP. To understand the structural basis of this new interaction, the authors solved two co-crystal structures of MANF in complex with BiP NBD. Lastly, the authors showed that MANF attenuates nucleotide exchange on BiP. Overall, this is an interesting study and for sure contributes to our understanding of ER quality control and protein homeostasis. However, it seems this study only scratched the surface of this interesting function of MANF and there are a number of points that the author need to address:

1. It is interesting that MANF interacts BiP's NBD and attenuates the nucleotide exchange on BiP. However, this study didn't really address mechanism. How does this interaction affect BiP's activity in protein folding, quality control, and unfolded protein response in ER? The authors attempted to address this fundamental question at the very end of the result section. They showed more ER proteins were in detergent insoluble high molecular weight chaperone-client complexes in MANF knockout cells than WT cells and suggested that this result supports the role of MANF in stabilizing certain BiP-client complexes. However, this seems a very vague experiment.

In the discussion, the authors compared MANF's NEI activity and role with those of Hip. "Hip is believed to facilitate the handover of clients from Hsp70 to Hsp90, for maturation, or to CHIP, for destruction". In Figure 2, the authors showed that MANF was crosslinked to BiP, Grp94 and Grp170, which seems to support "MANF may stabilize certain BiP-client complexes so as to promote their transfer to specific downstream quality control effectors." Testing analogous role of MANF in this aspect will increase the impact of this work and improve the manuscript for publication in Nature Communications.

2. It is a significant accomplishment that the authors obtained crystal structures of MANF in complex with BiP NBD. However, the authors did quite superficial comparison to previously published structures including Hip-Hsp70 NBD and various Hsp70-NEF. It seems the MANF-BiP interface is very different from all previously published NEI-Hsp70 and NEF-Hsp70 interfaces. Instead, it seems MANF-BiP interface overlaps with the Hsp70-J domain interfaces and the interface between NBD-SBD in the Hsc70 nucleotide-free structure (Jiang et al. Molecular Cell (2005) 20:513-524). These structural comparisons are important to understand the function and activity of MANF-BiP interaction.

3. How important is MANF for cellular function? What is the growth phenotype of MANF knockout cells? The authors showed MANF affects the nucleotide release kinetics of BiP close to 3 fold by itself and ~ 2 fold in the presence of Grp170. Will this modest effect have significant impact on BiP's function?

4. About the reported three structures, the R factors are high for the reported resolutions. This could be due to high Rmerge (especially for P1), low I/ σ for the high resolution shells and overall low redundancy of the data. They should improve these datasets and refinement before publication.

5. The organization of the manuscript needs to be improved for clarity. For example, the structures that they solved are complexes of MANF with BiP NBD. They should state that instead of "Structure of the BiP-MANF complex".

6. In Figure 4d, it seems all the mutations have similar $K_{1/2max}$ although the signals are different. Can the authors deduce the $K_{1/2max}$ for all the mutations, and then compare with WT MANF? In Figure 2e, the signals are different, but the $K_{1/2max}$ are quite similar.

Reviewer #2 (Remarks to the Author):

In the submitted manuscript, Ron and colleagues present structural and functional data focused on the previously described secreted neuroprotectant MANF. The authors show that MANF protein is present in cell lysates, and that this internal pool of MANF can be isolated from cross-linked cell lysates in complex with the ER Hsp70 BiP. They find that the SAP domain of MANF can complex with the nucleotide-binding domain of BiP, and structural data reveal that the SAP domain associates with a BiP cleft that has previously been identified as a binding site for the BiP interdomain linker in a BiP -ATP / BiP open state. The authors further demonstrate that SAP can antagonize nucleotide exchange BiP, and that less BiP is found in high molecular weight complexes in cells lacking MANF. Together these data lead the authors to propose that MANF may act similarly in cells to the cytoplasmic Hsp70 co-factor Hip, acting to dampen nucleotide exchange and stabilized BiP-client protein complexes. Overall this is a sound manuscript highlighting a potential new BiP co-factor within the ER. A weakness is that the paper relies primarily on in vitro evidence for the proposed role for MANF as a NEI. However, how a NEI might impact BiP function in cells is hard to address experimentally, and the presented data from the MANF^{-/-} cells are consistent with the current proposed model.

Specific comments:

1. The manuscript title introduces the alternate term ARMET for MANF. This name is not ever mentioned within the manuscript text (other than reference titles). If this is a standard alternative name, it would be worth mentioning also in the abstract / introduction.

2. The authors conclude that SAP is able to inhibit nucleotide (MABA-ADP) release in the presence of Grp170 (Fig. 5d), and the authors conclude that this represents "independent action" of these two regulators. Yet it is unclear to what extent Grp170 is active in this assay (?). The k_{off} in the presence of Grp170/ATP is reported to fall within 0.11-0.15 sec⁻¹ relative to the ~0.09 sec⁻¹ in the panels b and c). In addition, the 200 uM MANF used in the assay is in vast excess of the 1.25 uM Grp170.

Could the authors please elaborate on the activity of the Grp170 prep and also on the choice of concentrations for Grp170/MANF. Are these ratios based on relative affinities? on inhibitory / stimulatory values?

3. Related to point 2, it would be helpful if the authors would convert their bar graphs to instead show individual data points / some type of dotplot.

4. The inclusion of data with Grp170, and the inclusion of Grp94 in the model figure, hark back to the Figure 2a that showed isolation of Grp170 and Grp94 in the FLAG-MANF pulldown. Yet this is never explicitly discussed. I would suggest a sentence or two in the discussion speculating at to why both of the proteins were pulled down by MANF (e.g. ternary complex of BiP-Grp170-MANF?).

Reviewer #3 (Remarks to the Author):

Yan, Rato et al. present a study that provides detailed structural and functional insights into the intracellular role of MANF in protein folding. Moreover, the interaction of MANF with BiP and its role in stabilizing the ADP-bound form of BiP is clearly and analytically established. In general, the manuscript is well presented and organized. Briefly, it is shown that BiP can be recovered from lysate with FLAG-MANF IPs, thus confirming earlier findings from the literature. In a series of BLI assays, the authors show that it is the NBD domain of BiP that interacts with the SAP domain of MANF. Noteworthy is the alteration in the binding signal depending on which nucleotide BiP binds – ATP or ADP. The aforementioned findings were confirmed by co-crystallization experiments that underline steric differences in the interaction between the two proteins caused by the binding state confirmation of the NBD being ADP or ATP. The authors further confirm that MANF stabilizes BiP-ADP via a fluorescence assay. Furthermore, the authors use quantitative mass spectrometry in order to underline the biological impact and consistency of their findings. This experiment, in particular, is not only an interesting addition in terms of biological implication but also adds a further methodology level to the manuscript. Nevertheless, I would like to take the liberty to make some remarks regarding a set of items in the manuscript:

Quantitative proteomic study

1) Figure 6c top: Should not the peptide ratio/MS signals of the MANF k.o. and the wt be either 1:1 or behave in a similar pattern? This question comes from the fact that on the left panel MANF k.o. peptide shows a higher intensity compared to wt, whereas on the right MANF k.o. presents a lower intensity compared to wt. In comparison, the lower panel shows the expected peptide behaviour. Compare to Supplementary Figure 5b, here all peptides behave as expected.

2) Figure 6d: The majority of the peptides mapped across BiP in the input (left panel) show similar quantitative results. However, in the HMW complex (right panel) this is not true for all identified peptides. For example, two brackets around amino acid position 200 of BiP indicate the presence of two identified peptides. Since, they cover approximately the same part of the protein sequence it would be expected that the quantitative results are the same. However, both have quite different quantitative values with 0.87 and 0.64. A similar behaviour can be observed for two peptides of the SBD region (amino acid position approx. 620).

Do the authors have an explanation why some peptides/parts of the protein seem to show strong quantitative differences in this experiment? Are these peptides e.g. modified? May the authors also refer to Supplementary Figure 5c. Here, differences in quantification can be observed as well. Interestingly again similar regions of the protein. It might be interesting to elaborate if this region is of special interest from a structural point of view.

3) Lines 474 ff, based on which parameter was this normalisation for the different steps performed?

4) Line 198, please reference the SILAC paper (Ref. 45 manuscript) also here.

5) The MS parameter part could use some further details e.g. instrument parameter, fixed and variable modifications.

6) As the crystal structures have been made publicly available, it would be appreciated if the data of the MS experiment would be made available as well.

Further comments:

1) Supplementary Figure 1b: Please remove black line over the orange bar in the lower panel.

2) Supplementary Figure 3: May provide higher resolution image.

Overall, the study by Yan, Rato et al. is performed very well, the drawn conclusions from the depicted experimental results is consistent and it was a pleasure to read.

Point-by-point response to the critiques:

(The original critiques are quoted in black ink and our response provided in ochre ink)

Reviewer 1:

1. It is interesting that MANF interacts BiP's NBD and attenuates the nucleotide exchange on BiP. However, this study didn't really address mechanism. How does this interaction affect BiP's activity in protein folding, quality control, and unfolded protein response in ER? The authors attempted to address this fundamental question at the very end of the result section. They showed more ER proteins were in detergent insoluble high molecular weight chaperone-client complexes in MANF knockout cells than WT cells and suggested that this result supports the role of MANF in stabilizing certain BiP-client complexes. However, this seems a very vague experiment.

In the discussion, the authors compared MANF's NEI activity and role with those of Hip. "Hip is believed to facilitate the handover of clients from Hsp70 to Hsp90, for maturation, or to CHIP, for destruction". In Figure 2, the authors showed that MANF was crosslinked to BiP, Grp94 and Grp170, which seems to support "MANF may stabilize certain BiP-client complexes so as to promote their transfer to specific downstream quality control effectors." Testing analogous role of MANF in this aspect will increase the impact of this work and improve the manuscript for publication in Nature Communications.

To further build confidence in the observation that fewer molecules of BiP associate with unfolded clients in MANF knockout cells (a conclusion based on the SILAC experiment, shown in Fig. 6), we examined the case of a specific misfolded protein - a truncated version of the secreted SERPIN, alpha-1 antitrypsin (null Hong Kong, A1AT-NHK). New supplemental Fig. 5d shows that about half as much BiP is recovered in complex with A1AT-NHK in MANF knockout cells, compared with wildtype cells.

2. It is a significant accomplishment that the authors obtained crystal structures of MANF in complex with BiP NBD. However, the authors did quite superficial comparison to previously published structures including Hip-Hsp70 NBD and various Hsp70-NEF. It seems the MANF-BiP interface is very different from all previously published NEI-Hsp70 and NEF-Hsp70 interfaces. Instead, it seems MANF-BiP interface overlaps with the Hsp70-J domain interfaces and the interface between NBD-SBD in the Hsc70 nucleotide-free structure (Jiang et al. Molecular Cell (2005) 20:513-524). These structural comparisons are important to understand the function and activity of MANF-BiP interaction.

In the revised manuscript, the consequences of the engagement of MANF's SAP domain with BiP's NBP have been further clarified by means of a video in which BiP is shown to "morph" from its domain undocked (ADP bound) state that can accommodate MANF binding to its domain docked (ATP bound) state in which clashes with a bound molecule of MANF are noted. We believe new supplemental video 1 provides a plausible structural explanation for MANF's action as an NEI and showcases the important differences by which this activity arises in MANF and Hip.

The reviewer's comment also drew attention to an important potential consequence of the fact that MANF binds BiP on the same surface contacted by J-domain proteins. It thus seemed important to address the possibility that MANF has ATPase stimulatory activity;

such activity would not have been revealed by the assays presented in the original version of the manuscript. New supplemental Fig. 3b, compares the ATPase activity of BiP (using a single turnover assay in which the conversion of ^{32}P alpha-labelled ATP to ADP is monitored by thin layer chromatography) in the presence of J-domain, which greatly accelerates it (as expected) and MANF, which, we find has no significant effect.

3. How important is MANF for cellular function? What is the growth phenotype of MANF knockout cells? The authors showed MANF affects the nucleotide release kinetics of BiP close to 3 fold by itself and ~ 2 fold in the presence of Grp170. Will this modest effect have significant impact on BiP's function?

The reviewer questions the significance of the nucleotide exchange inhibitory (NEI) activity of MANF in the presence of a nucleotide exchange factor (NEF), pointing to the paltry increase in nucleotide exchange induced by the NEF Grp170 (Fig. 5d in the original manuscript). We have discovered that inclusion of a physiological concentration of calcium (0.5 mM) enforces an exchange regime in which the NEF activity of Grp170 is conspicuous. New Fig. 5d shows that MANF exhibits marked NEI activity in this physiological context, counteracting the effects of Grp170. We thank the reviewers for catalysing this clarifying modification to this experiment.

4. About the reported three structures, the R factors are high for the reported resolutions. This could be due to high Rmerge (especially for P1), low I/σ for the high resolution shells and overall low redundancy of the data. They should improve these datasets and refinement before publication.

In response to the referee's concern about the values of the R-factors for the crystal structures, we re-checked the three datasets using phenix.Xtriage. No obvious problems such as twinning, tNCS, ice rings or significant anisotropy were identified. We also ran comprehensive validation of the three refined structures in Phenix. The R_{free} and R_{work} of the NBD-MANF and NBD-SAP complexes are typical for their resolution according to the Polygon tool and the percentile scores in the PDB validation reports. These R values are not the best among structures at similar resolutions, probably because of the flexibility of the MANF's SAP domain as shown in the crystal structure of MANF (PDB 2W51) and NMR structures (PDB 2KVD, 2RQY).

We tried further refinement using the previously deposited BiP-V461F dataset, because the R_{free} was indeed in the higher end of the range seen for structures at similar resolution. However, the R_{work} and R_{free} values did not improve significantly. Following this, we performed a series of paired refinements with resolution cutoffs in the range from 2.08 Å to 2.6 Å according to the method published by Karplus and Diederichs (PMID: 22628654) and compared R values computed with data limited to 2.6 Å resolution, common to all the refinements. The R_{work} and R_{free} of the model refined to 2.6 Å are in the middle of the range seen in structures determined at similar resolution limits. As shown in the figure below, adding data to higher resolution reduces R_{free} for the data to 2.6 Å (indicating improvement in the structure) while increasing R_{work} (indicating less overfitting); the reduction in the gap between R_{free} and R_{work} also shows that overfitting has been reduced. $CC_{1/2}$ is 0.602 in the high-resolution (2.08 Å) resolution shell, which is higher than the threshold of 0.3 suggested by Karplus and Diederichs, and the $\langle I/\sigma \rangle$ at 1.4 is not far below a traditional cutoff of 2.0. As pointed out by Karplus and Diederichs, R_{merge} values can rise to values much higher than what has traditionally been thought of as allowable (~ 0.6) by using the $CC_{1/2} > 0.3$ criteria.

Therefore, we retained the original resolution limit of 2.08 Å and re-submitted the re-refined model, which we believe to be of good quality despite the relatively high R values.

A paired refinement demonstrates that the extended resolution improves the quality of the model. Plotted on the left axis are R_{free} (black line, solid circle), R_{work} , (grey line, solid square), and plotted on the right axis is $R_{\text{free}}-R_{\text{work}}$ (dotted line, open triangle) calculated at 2.6 Å for paired refinements in which the model was first refined against data limited to 2.08, 2.2, 2.4, or 2.6 Å. The chosen resolution of 2.08 Å shows a decrease in R_{free} and an increase in R_{work} , which proves that using the extra resolution improves the model while reducing overfitting.

5. The organization of the manuscript needs to be improved for clarity. For example, the structures that they solved are complexes of MANF with BiP NBD. They should state that instead of “Structure of the BiP-MANF complex”.

The revised version specifically mentions that the complex whose structure has been solved is that of SAP-NBD and MANF-NBD; a BiP-MANF complex is not mentioned.

6. In Figure 4d, it seems all the mutations have similar $K_{1/2\text{max}}$ although the signals are different. Can the authors deduce the $K_{1/2\text{max}}$ for all the mutations, and then compare with WT MANF? In Figure 2e, the signals are different, but the $K_{1/2\text{max}}$ are quite similar.

In the revised manuscript we have clarified that whilst the association of MANF (and its weaker mutants) with BiP-NBD gives rise to a plot of saturable binding from which a $K_{1/2}$ max value can be extracted, the association of the stronger mutants is too feeble to extract a $K_{1/2}$ max. The reliable $K_{1/2}$ max values are presented in the revised version of Fig. 4c and the absence of meaningful association of the strong mutants of MANF with BiP-NBD is likewise indicated. We thank the reviewer for bringing this to our attention.

Reviewer 2:

1. The manuscript title introduces the alternate term ARMET for MANF. This name is not ever mentioned within the manuscript text (other than reference titles). If this is a standard alternative name, it would be worth mentioning also in the abstract / introduction.

As MANF is the broadly accepted term for this protein and the one used in most publications, we have removed ARMET from the title.

2. The authors conclude that SAP is able to inhibit nucleotide (MABA-ADP) release in the presence of Grp170 (Fig. 5d), and the authors conclude that this represents "independent action" of these two regulators. Yet it is unclear to what extent Grp170 is active in this assay (?). The k_{off} in the presence of Grp170/ATP is reported to fall within 0.11-0.15 sec⁻¹ relative to the ~0.09 sec⁻¹ in the panels b and c). In addition, the 200 uM MANF used in the assay is in vast excess of the 1.25 uM Grp170.

Could the authors please elaborate on the activity of the Grp170 prep and also on the choice of concentrations for Grp170/MANF. Are these ratios based on relative affinities? on inhibitory / stimulatory values?

(Please see response to point 3 of reviewer 1)

3. Related to point 2, it would be helpful if the authors would convert their bar graphs to instead show individual data points / some type of dotplot.

The bar graphs in the revised manuscript now include the dotplot, showing all the data points from all the replicates.

4. The inclusion of data with Grp170, and the inclusion of Grp94 in the model figure, hark back to the Figure 2a that showed isolation of Grp170 and Grp94 in the FLAG-MANF pulldown. Yet this is never explicitly discussed. I would suggest a sentence or two in the discussion speculating at to why both of the proteins were pulled down by MANF (e.g. ternary complex of BiP-Grp170-MANF?).

The last paragraph of the discussion in the revised manuscript now harkens to this point: *"ER chaperones function in physically-linked networks^{40,41}, to which MANF may connect, as possibly hinted by the recovery of both Grp94 and the NEF Grp170 in the MANF pull down (Fig. 2a)"*

Reviewer 3:

1) Figure 6c top: Should not the peptide ratio/MS signals of the MANF k.o. and the wt be either 1:1 or behave in a similar pattern? This question comes from the fact that on the left panel MANF k.o. peptide shows a higher intensity compared to wt, whereas on the right MANF k.o. presents a lower intensity compared to wt. In comparison, the lower panel shows the expected peptide behaviour. Compare to Supplementary Figure 5b, here all peptides behave as expected.

As the reviewer is undoubtedly aware, the ratiometric nature of SILAC empowers it to overcome the distorting consequences of subtle differences in the concentration of constituents (BiP, in this case) in the two samples being analysed. This is now emphasized in the description of the results in the revised manuscript.

2) Figure 6d: The majority of the peptides mapped across BiP in the input (left panel) show similar quantitative results. However, in the HMW complex (right panel) this is not true for all identified peptides. For example, two brackets around amino acid position 200 of BiP indicate the presence of two identified peptides. Since, they cover approximately the same part of the protein sequence it would be expected that the quantitative results are the same. However, both have quite different quantitative values with 0.87 and 0.64. A similar behaviour can be observed for two peptides of the SBD region (amino acid position approx. 620).

Do the authors have an explanation why some peptides/parts of the protein seem to show strong quantitative differences in this experiment? Are these peptides e.g. modified? May the authors also refer to Supplementary Figure 5c. Here, differences in quantification can be observed as well. Interestingly again similar regions of the protein. It might be interesting to elaborate if this region is of special interest from a structural point of view.

We thank the perspicacious reviewer for flagging this point. In response we have scrutinized the peptides that exhibit greater variance and have been unable to come up with a unifying feature distinguishing them from peptides with less variance. We have alerted the reader to this point in the revised manuscript, whilst noting that the existence of such variance does not alter the conclusion from this experiment: *“Whilst the source of the variation in the extent of the genotype-based biased recovery of some BiP peptides over others remains unknown, the bias itself was a consistent finding, observed in SILAC experiments with heavy isotopes marking either genotype and over all 18 detectable BiP tryptic peptides (Fig. 6d)”*.

3) Lines 474 ff, based on which parameter was this normalisation for the different steps performed?

In the revised manuscript we detail that the parameter used was protein concentration assessed by the Bio-Rad protein assay reagent.

4) Line 198, please reference the SILAC paper (Ref. 45 manuscript) also here.

Done

5) The MS parameter part could use some further details e.g. instrument parameter, fixed and variable modifications.

Done; see revised methods section

6) As the crystal structures have been made publicly available, it would be appreciated if the data of the MS experiment would be made available as well.

The raw data is now included in the source data file, submitted alongside the revised manuscript.

Further comments:

1) Supplementary Figure 1b: Please remove black line over the orange bar in the lower panel.

Done

2) Supplementary Figure 3: May provide higher resolution image.

Higher resolution image provided

Reviewers' comments:

Reviewer #1 (Remarks to the Author):

I appreciate the authors' efforts in addressing my concerns. However, I feel the authors should address my remaining concerns before publication:

1. For the first point from my previous review, besides the in vivo assay with MANF knockout cells, is it possible for the authors to provide some direct biochemical supports such as refolding or preventing protein aggregation using BiP to explore the mechanism of MANF's activity? The authors are experts in the chaperone field. I feel this will increase the impact of this work unless such assays turn out to be intractable.
2. I appreciate the authors' effort in addressing the second point of my previous review using a single turnover assay. However, the results are of poor quality. This assay has been used for a number of Hsp70s and is the gold standard for assaying the ATPase activity of Hsp70s. There are a number of issues with new Supplemental Fig. 3b: 1) at 0 time, the majority of ATP has already been hydrolyzed, indicating either the assay was not done appropriately or there might be contaminants in their purified BiP protein; 2) it seems the ATPase activity of their BiP is much higher than all the published ATPase activity for Hsp70s including BiP, again suggesting there might be contaminants in their purified BiP protein; 3) the J-domain of ERdj6 was used as a positive control; however, previously published data suggested that the J-domain alone is not sufficient to stimulate the ATPase activity of Hsp70s.
3. For the third point from my previous review, I'm confused. It seems to me the new Fig. 5d still showed that MANF's effect on BiP is about 2-3 fold with or without Grp170.
4. For the fourth point of my previous review, I applaud the authors' effort.
5. For the fifth point of my previous review, the title of the second section of the Results is still "Structure of the BiP-MANF complex" and I feel this is misleading.
6. For the new Fig. 4d, the R2s are quite poor.

Reviewer #2 (Remarks to the Author):

The additions to the manuscript satisfy my initial concerns.

Reviewer #3 (Remarks to the Author):

The revised manuscript addresses all previously raised points. Unfortunately, the authors were unable to find a property that explains the quantitative abnormality in the SILAC data for some peptides. However, the added statement is sufficient, especially in respect of the scope of this excellent manuscript.

We have considered carefully the comments of reviewer 1 and have modified the manuscript in response. Below is a point-by-point discussion of the reviewer's comments and our response to them

1. For the first point from my previous review, besides the *in vivo* assay with MANF knockout cells, is it possible for the authors to provide some direct biochemical supports such as refolding or preventing protein aggregation using BiP to explore the mechanism of MANF's activity? The authors are experts in the chaperone field. I feel this will increase the impact of this work unless such assays turn out to be intractable.

Point 1 is a request to *provide some direct biochemical supports...* the reviewer does not suggest a critical and doable experiment to test a specific hypothesis or challenge our claim that MANF inhibits nucleotide exchange on BiP. That said, we believe, the manuscript has a finding that relates to the reviewer's concern, one that was perhaps under-emphasized in previous versions: Figure 5g & 5h show that MANF-mediated inhibition of nucleotide exchange retards ATP-mediated client release from ADP-bound BiP. Page 8 of the revised manuscript emphasizes the significance of this reductionist *in vitro* experiment to the conclusion of our paper.

2. I appreciate the authors' effort in addressing the second point of my previous review using a single turnover assay. However, the results are of poor quality. This assay has been used for a number of Hsp70s and is the gold standard for assaying the ATPase activity of Hsp70s. There are a number of issues with new Supplemental Fig. 3b: 1) at 0 time, the majority of ATP has already been hydrolyzed, indicating either the assay was not done appropriately or there might be contaminants in their purified BiP protein; 2) it seems the ATPase activity of their BiP is much higher than all the published ATPase activity for Hsp70s including BiP, again suggesting there might be contaminants in their purified BiP protein; 3) the J-domain of ERdj6 was used as a positive control; however, previously published data suggested that the J-domain alone is not sufficient to stimulate the ATPase activity of Hsp70s.

The reviewer seems unhappy with the ADP signal found at $t = 0$ of the time course experiments in Supplemental Fig. 3b. In the legend to the revised version of supplemental figure 3b, we now explain that the amount of ADP present at $t = 0$ in a single turnover experiment is immaterial to its interpretation, as the (relative) rates of ATP hydrolysis (by BiP) are compared by the decrease in ATP at the various time points, over the conditions present at $t=0$ in each experimental series. By this criterion, the experiment clearly shows that the J-domain of ERdj6 is able to stimulate the hydrolysis rate of ATP by BiP whilst MANF is unable to do so. We go on to note that the ADP signal present at $t = 0$ arises from a combination of factors: non-enzymatic hydrolysis of the (unlabelled) γ phosphate during storage of the precursor ^{32}P α -labelled ATP, enzymatic hydrolysis during formation of the BiP-ATP complexes and possibly hydrolysis that occurs during sample freezing and thawing. However, as this is a single turnover experiment (performed in the absence of free nucleotide) only pre-formed BiP-ATP complexes can hydrolyse ATP. Thus, pre-existing BiP-ADP complexes are inert and the pre-experimental conversion of ATP to ADP is ignored. In other words only the ATPase activity of BiP-ATP complex during the experiment are taken into account. Regarding the purity of our BiP preparations and the potential presence of a contaminating ATPase we note, that ATPase-inactive BiP (BiP^{T229A}) prepared by the same method has no ATPase activity.

The provenance of ADP present at $t = 0$ may also bear on reviewer's concern that this assay reported on an expectedly-high ATPase activity of BiP. But given the multiple factors that contribute to the amount of ADP present at $t = 0$ is not a helpful clue to BiP's ATPase activity. It may also be edifying to consider that using the same assay, we have previously established that the basal ATPase rate of BiP is slower by factor 5-10 than nucleotide exchange (ADP release) and thus is the rate-limiting step of BiP's ATPase cycle¹, which is consistent with other Hsp70s such as DnaK². The basal ATPase rate of other Hsp70s is in the order of 10^{-3} - 10^{-4} per second and we measured the rate for BiP 0.002 s^{-1} (¹), which is very similar to one reported previously for BiP 0.003 s^{-1} (³). Furthermore, the stimulatory effect of J6 is HPD-motif dependent¹. Thus, we maintain that the assay used here is valid and that our conclusion that MANF does not accelerate BiP's ATPase activity stands.

We take issue with the reviewer's claim that "previously published data suggested that the J-domain alone is not sufficient to stimulate the ATPase activity of Hsp70s. Hsp70 co-chaperones are bipartite with separate client protein-targeting domains and ATPase-stimulatory J-domains. It is well established that the J-domain is capable of stimulating the basal ATPase activity of Hsp70s in absence of additional substrate or the co-chaperone's substrate binding domain. This was first demonstrated in 1994 by the Georgopoulos lab who noted that the isolated J domain of *E. coli* DnaJ stimulated the ATPase activity of DnaK in vitro⁴ (figure 3 therein) and has since been observed by others, including ourselves as applied to the isolated J domain of ERDJ6 and BiP¹ (figure 6, therein). Thus, the use of the isolated J domain is fully justified by experimental expediency.

3. For the third point from my preview review, I'm confused. It seems to me the new Fig. 5d still showed that MANF's effect on BiP is about 2-3 fold with or without Grp170.

We find that MANF inhibits nucleotide exchange on BiP by 2-3 fold. MANF does so both in the absence of calcium and nucleotide exchange factor (Fig. 5b & 5c) or under more physiological conditions, in the presence of calcium and nucleotide exchange factor (Fig 5d). Two to three fold less, is significant, when it comes to nucleotide exchange. Consider that MANF antagonizes more than half the increase in nucleotide exchange brought about by the NEF Grp170 (compare the first three bars from the left in figure 5d). We have added a sentence highlighting this point to the revised manuscript (page 8).

4. For the fourth point of my previous review, I applaud the authors' effort.

We take this to mean that the reviewer is satisfied by our response to their critique.

5. For the fifth point of my previous review, the title of the second section of the Results is still "Structure of the BiP-MANF complex" and I feel this is misleading.

Though the crystal structures we present are those of complexes between MANF (or its SAP domain) and the NBD of BiP, we believe that the insights gained are relevant to the complex formed by the intact proteins. For years, researchers have legitimately inferred the details by which two proteins interact from structural data derived from individual domains and even fragments of proteins, either in complex or in isolation. Therefore, we do not believe we are misleading our readers with the title as it currently stands. But as it appears the reviewer has strong views on the matter we have (reluctantly) agreed to modify the title to read: "Structural insights into the BiP-MANF complex".

6. For the new Fig. 4d, the R2s are quite poor.

The reviewer is not challenging our conclusion that the mutations compromise MANF binding to BiP. We believe that R^2 of 0.966, 0.978 and 0.913 (the fits of the data derived from the binding of WT, R23A and K138A MANF to BiP) are an indication of the reliability of the data. We further believe that the inability to fit binding data to a model when there is no measurable binding (R133E and E153A) is likewise reliable. We have modified the text in the figure legend to explicitly render these points.

1. Preissler, S. et al. AMPylation targets the rate-limiting step of BiP's ATPase cycle for its functional inactivation. *Elife* **6**, e29428 (2017).
2. Theyssen, H., Schuster, H.P., Packschies, L., Bukau, B. & Reinstein, J. The second step of ATP binding to DnaK induces peptide release. *J Mol Biol* **263**, 657-70 (1996).
3. Mayer, M., Reinstein, J. & Buchner, J. Modulation of the ATPase cycle of BiP by peptides and proteins. *J Mol Biol* **330**, 137-44 (2003).
4. Wall, D., Zylicz, M. & Georgopoulos, C. The NH₂-terminal 108 amino acids of the Escherichia coli DnaJ protein stimulate the ATPase activity of DnaK and are sufficient for lambda replication. *J Biol Chem* **269**, 5446-51 (1994).